# Non-coding cancer driver candidates identified with a sample- and position-specific model of the somatic mutation rate

Malene Juul[1]*, Johanna Bertl[1], Qianyun Guo[2], Morten Muhlig Nielsen[1], Michał Świtnicki[1], Henrik Hornshøj[1], Tobias Madsen[1], Asger Hobolth[2], Jakob Skou Pedersen[1,2]*

[1]Department of Molecular Medicine (MOMA), Aarhus University Hospital, Aarhus, Denmark; [2]Bioinformatics Research Centre (BiRC), Aarhus University, Aarhus, Denmark

**Abstract** Non-coding mutations may drive cancer development. Statistical detection of non-coding driver regions is challenged by a varying mutation rate and uncertainty of functional impact. Here, we develop a statistically founded non-coding driver-detection method, ncdDetect, which includes sample-specific mutational signatures, long-range mutation rate variation, and position-specific impact measures. Using ncdDetect, we screened non-coding regulatory regions of protein-coding genes across a pan-cancer set of whole-genomes (n = 505), which top-ranked known drivers and identified new candidates. For individual candidates, presence of non-coding mutations associates with altered expression or decreased patient survival across an independent pan-cancer sample set (n = 5454). This includes an antigen-presenting gene (*CD1A*), where 5'UTR mutations correlate significantly with decreased survival in melanoma. Additionally, mutations in a base-excision-repair gene (*SMUG1*) correlate with a C-to-T mutational-signature. Overall, we find that a rich model of mutational heterogeneity facilitates non-coding driver identification and integrative analysis points to candidates of potential clinical relevance.

*For correspondence: malene.juul.rasmussen@clin.au.dk (MJ); jakob.skou@clin.au.dk (JSP)

**Competing interests:** The authors declare that no competing interests exist.

## Introduction

Cancer is caused by somatically acquired changes in the DNA sequence of genomes (*Stratton et al., 2009*). Recently, large-scale sequencing of cancer-genomes coordinated by the International Cancer Genome Consortium (ICGC), The Cancer Genome Atlas (TCGA), and others has catalogued the molecular changes across hundreds of cancer samples (*Hudson et al., 2010*; *Weinstein et al., 2013*). The quest is now to analyze and understand the role of these changes in cancer development. The aberrations in non-coding regions are of particular interest as they have only become evident with the advent of whole cancer-genomes. Here, we develop the method ncdDetect for non-coding cancer driver detection. The method captures the heterogeneities of the mutational processes in cancer and aggregates signals of mutational burden as well as functional impact in the significance evaluation of a candidate driver element. We apply the method to 505 TCGA whole-genomes (*Fredriksson et al., 2014*).

Cancer arises by an evolutionary process where natural selection operates on genetic variation stemming from randomly occurring somatic mutations. Thousands of somatic mutations distinguish tumor tissue from healthy tissue, as a result of the mutational processes that cancer cells go through during the lifetime of a cancer patient. The somatic mutations are identified through Next-

**eLife digest** Cancers are diseases caused by changes in DNA sequences. Some changes occur in the protein-coding part of the DNA sequence, in other words, in the stretches of DNA that include the instructions to make a given protein. Other changes occur in the remaining parts of the DNA that do not code for proteins, which accounts for about 98% of the human genome. Modern technologies allow us to identify these DNA changes, but, up until recently, this has only been possible for the protein-coding part of the DNA. Many studies have thus analyzed DNA changes in the protein-coding part of the human genome, while the larger, non-coding part remains rather unexplored.

Advances in technology means that large datasets are becoming available where changes in DNA sequences are identified across the entire genomes of a collection of cancer patients. However, it is not clear which of these DNA changes play a role in the development of cancer and which are neutral with no effect on cancer.

Now, Juul et al. have developed a new method, named 'ncdDetect', to search the human genome and identify stretches of DNA that when changed give cancer cells an advantage and allow them to grow. Juul et al. refer to these DNA stretches as 'driver elements', and, after analyzing the genomes from 505 patients with cancer, they identified some known driver elements and some potentially new ones. For example, possible driver elements were found in non-coding parts of the DNA that regulate genes called *SMUG1* and *CD1A*. Both of these genes encode proteins that had been linked to cancer in the past, but driver elements had not previously been described in the nearby non-coding regions. Juul et al. also found a number of possible driver elements that might be important to consider in the treatment of cancers.

Importantly, not all the candidate driver elements identified with ncdDetect are true drivers. The changes in DNA vary greatly between different types of cancer and even between different cases of a single type of cancer. Understanding and describing this variation continues to be a challenge in identifying driver elements, and so Juul et al. plan to keep improving the method to make sure that the driver elements it identifies are all trustworthy.

Generation Sequencing (NGS) and commonly labeled according to their effect on cancer development: *Driver mutations* are subject to positive selection during the evolutionary process of cancer, as they offer the cell a growth advantage and contribute to the expansion of tumors. By definition, driver genes contain one or more driver mutations. *Passenger mutations*, on the other hand, have a neutral, or perhaps slightly negative, fitness contribution to the cell, and accumulate as passive passengers during the evolutionary process of cancer (*Stratton et al., 2009*; *Pon and Marra, 2015*). Many more passenger than driver mutations exist in cancer cells, and distinguishing the two is challenging (*Marx, 2014*). Typically, the strategy is to search for signs of recurrent positive selection across a set of cancer genomes.

Signs of recurrent positive selection across cancer genomes can be detected by comparing the somatic mutation frequency to an estimated background mutation rate (*Pon and Marra, 2015*). However, modeling the background mutation rate is complicated as it varies along the genome with a large degree of heterogeneity (*Lawrence et al., 2013*; *Polak et al., 2015*; *Alexandrov et al., 2013*). Not only does the mutation rate in cancer exhibit high variation between different cancer types; this is also the case between different samples of the same cancer type. Furthermore, the mutational processes are affected by various genomic features, primarily replication timing, expression, and the position-specific sequence context (*Lawrence et al., 2013*; *Bertl et al., 2017*). It is thus crucial to take these features into account when estimating the background mutation rate in cancer. Another strategy for detecting signs of positive selection in cancer is to rank mutations according to their impact on protein function (*Marx, 2014*). In particular, point mutations might introduce alterations in the amino acid sequence of a protein, and thereby change its function (*Reva et al., 2011*).

Systematic mutational screens of cancer exomes have expanded the set of known cancer driver genes over the past decade (*Forbes et al., 2015*). Many tools exist for the identification of such

genes (*Dees et al., 2012*; *Lawrence et al., 2013*; *Gonzalez-Perez and Lopez-Bigas, 2012*; *Tamborero et al., 2013*; *Reimand and Bader, 2013*), and at present, 616 cancer driver genes are catalogued as causally implicated in cancer (*Cosmic, 2016*). However, less than 2% of the genome codes for protein. While it is established that non-coding elements play diverse roles in regulating the expression of protein-coding genes, only few studies systematically explored the role of non-coding somatic mutations in cancer development (*Weinhold et al., 2014*; *Lochovsky et al., 2015*; *Melton et al., 2015*; *Fredriksson et al., 2014*; *Mularoni et al., 2016*; *Lanzós et al., 2017*). The first identified non-coding driver element was the *TERT* promoter with highly recurrent mutations across several cancer types (*Huang et al., 2013*; *Horn et al., 2013*). In general, the functional understanding of non-coding regions is poor compared to protein-coding regions, challenging the interpretation of non-coding mutations (*Khurana et al., 2016*).

We develop the method ncdDetect for detection of non-coding cancer driver elements. With this method, we consider the frequency of mutations alongside their functional impact to reveal signs of recurrent positive selection across cancer genomes. In particular, the observed mutation frequency is compared to a sample- and position-specific background mutation rate, which is estimated based on various genomic annotations. A scoring scheme (e.g. position-specific evolutionary-conservation scores) is applied to further account for functional impact in the significance evaluation of a candidate cancer driver element.

To strengthen our conclusions regarding the driver potential of candidate elements, we draw on additional data sources. Non-coding mutations may perturb gene expression patterns, and we thus correlate their presence with expression levels in an independent data set (*Ding et al., 2015*). Likewise, we correlate mutation status with survival information for these candidates.

What sets ncdDetect aside from other non-coding driver detection methods is the position-specificity, and the derived ability to include genomic annotations of varying resolution down to the level of individual positions. In one existing non-coding driver detection method, the position- and sample-specific probabilities of mutation are derived, much like in ncdDetect but are then aggregated across a candidate element during significance evaluation (*Melton et al., 2015*). This means that knowledge about the exact position and probability of a mutation is not fully utilized. In another method, the genome is divided into bins according to the average value of replication timing (*Lochovsky et al., 2015*), and in a recent method, the significance evaluation is performed by locally conditioning on the number of observed mutations within a candidate element (*Mularoni et al., 2016*). To our knowledge, no existing non-coding driver detection method derive and apply position- and sample-specific probabilities of mutation in the significance evaluation of a candidate driver element, and allows the use of position-specific scores and accurate evaluation of their expectation across a candidate element. This unique feature of ncdDetect means that candidate elements of arbitrary size and location can be analyzed, and that the potential large variation of mutational probabilities within a candidate element is handled.

With ncdDetect, we model the different levels of heterogeneity in the somatic mutation rate known to be at play in cancer and evaluate the relative merit of different position-specific scoring-schemes. The result is a driver detection method tailored for the non-coding part of the genome, and with it we aim to contribute to the understanding of non-coding cancer driver elements.

## Results

ncdDetect evaluates if a given non-coding element is under recurrent positive selection across cancer samples. The method takes as input (a) a candidate genomic region of interest, (b) position- and sample-specific probabilities of mutation, and (c) position- and sample-specific scores measuring mutational burden or impact.

### Position- and sample-specific model of the background mutation rate

A key feature of ncdDetect is the application of position- and sample-specific probabilities of mutation. These are obtained by a statistical null model, inferred from somatic mutation calls of a collection of cancer samples (Material and methods: Statistical null model) (*Bertl et al., 2017*). The model predicts the mutation rate from a set of explanatory variables, that is genomic annotations (*Figure 1A*). In the present paper, the null model is trained on 505 whole genomes distributed across 14 different cancer types generated by TCGA (*Fredriksson et al., 2014*) (*Figure 1B*).

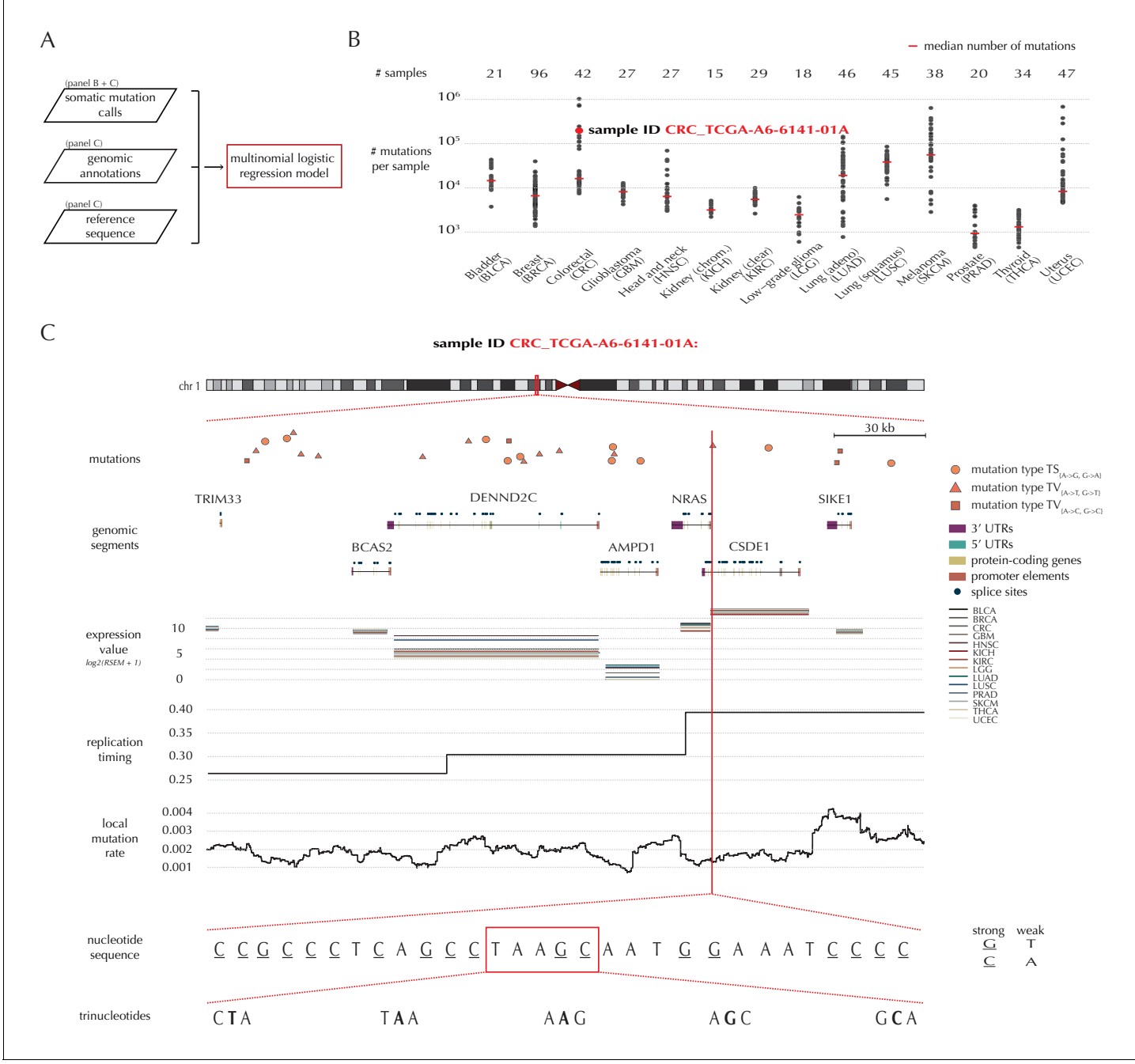

**Figure 1.** Variation in mutation rate at different scales and various explanatory variables. (**A**) The flowchart illustrates the input to the model fit that predicts the position- and sample-specific mutational probabilities. (**B**) The number of mutations observed per sample divided into the 14 different cancer types. (**C**) The set of genomic annotations used as explanatory variables are illustrated on a 300 kb region of chromosome 1 for the colorectal cancer sample CRC_TCGA-A6-6141-01A. For illustrative purposes, the nucleotide sequence is shown on a 30 bp section of chromosome 1 and trinucleotides likewise on a 5 bp section.

The following source data and figure supplements are available for figure 1:

**Source data 1.** The number of mutations observed for each of the 505 samples.

**Figure supplement 1.** The average number of mutations observed per sample per bp for each of the considered element types, as well as for intergenic regions.

*Figure 1 continued*

**Figure supplement 1—source data 1.** The number of mutations per sample per bp for the defined element types.

As explanatory variables, the model includes genomic annotations known to correlate with the mutation rate in cancer, as well as annotations we have found to improve the model fit. It is well known that the mutation rate varies between samples (*Lawrence et al., 2013*; *Alexandrov et al., 2013*). Indeed, the mean and median number of mutations per sample is approximately $32 \times 10^3$ and $8 \times 10^3$, respectively, with a large degree of variation between and within cancer types (*Figure 1B*). For example, the average number of mutations per sample is 73 times higher for colo-rectal cancer than for thyroid cancer, and within melanoma cancer, the least mutated sample has 224 times fewer mutations compared to the highest mutated sample. The mutation rate depends on the position-specific sequence context (*Alexandrov et al., 2013*) and correlates with replication tim-ing and gene expression level (*Lawrence et al., 2013*). The mutation rate also varies between differ-ent types of genomic regions (*Weinhold et al., 2014*). Finally, we find that the local mutation rate in a window around a given genomic position helps to capture unaccounted mutation rate variation and increases the goodness of fit. The complete model specification, including the definition of the local mutation rate, is given in Material and methods: Statistical null model. Consequently, genomic annotations considered as explanatory variables in the null model for each sample are *replication timing*, tissue-specific *gene expression level*, *trinucleotides* (the nucleotide under consideration and its left and right flanking bases, thus taking into account the sample-specific mutational signature), *genomic segment* (3' and 5' untranslated regions (UTRs), splice sites, promoter elements and pro-tein-coding genes) and *local mutation rate* (*Figure 1C*). Given these explanatory variables for a spe-cific genomic position, the model predicts the particular probability of observing a mutation of a given type for a specific sample at this particular position.

## Strand symmetric model
The reference sequence is divided into weak (A and T) and strong (G and C) base pairs (bps) (*Cor-nish-Bowden, 1985*). As strands generally cannot be distinguished in non-coding regions, we handle them symmetrically, with weak bps denoted with an A and strong bps denoted with a G. The 12 dif-ferent types of point mutations are thus collapsed into the six classes, 'A→C' (thus including both A→C and T→G mutations), 'A→G', 'A→T', 'G→T', 'G→C' and 'G→A'.

Our model considers four possible outcomes for each position: Transitions ($TS_{\{A\to G,\ G\to A\}}$), two types of transversions ($TV_{\{A\to T,\ G\to T\}}$ and $TV_{\{A\to C,\ G\to C\}}$) as well as the reference class of no mutation.

## Mutation-rate predictions and position-specific scores
For a given non-coding element of interest, the null model ensures the availability of position- and sample-specific probabilities of each of the four possible outcomes (*Figure 2A*). Due to the rarity of observing a mutation, the predicted mutational probabilities are of small magnitude (*Figure 2B–C*, *Figure 1—figure supplement 1*). Additional to these probabilities, each position is associated with a score that may be sample-specific, and may depend on the outcome class (*Figure 2D–E*). The scor-ing scheme can be freely defined, and in the present paper, we illustrate three different choices of scores, namely the number of mutations, log-likelihoods and conservation scores (*Figure 2E*). A wide variety of other scoring schemes can be considered. In particular, the flexibility of ncdDetect allows for different scoring schemes to be chosen for different types of candidate elements.

## Scoring schemes
A good scoring scheme must be able to discriminate well between events that constitute true cancer drivers and events that are neutral. The scoring scheme can, for example, evaluate the mutational burden and be defined by means of the number of mutations in a candidate region. This approach has been taken by existing non-coding cancer driver detection methods (*Lochovsky et al., 2015*). Here, the score for a given position is defined to be one if a mutation of any type occurs, and zero if no mutation occurs. Another approach is to evaluate the goodness of fit of the observed mutations to the null model, and define the scores as log-likelihoods, that is, minus the natural logarithm of the

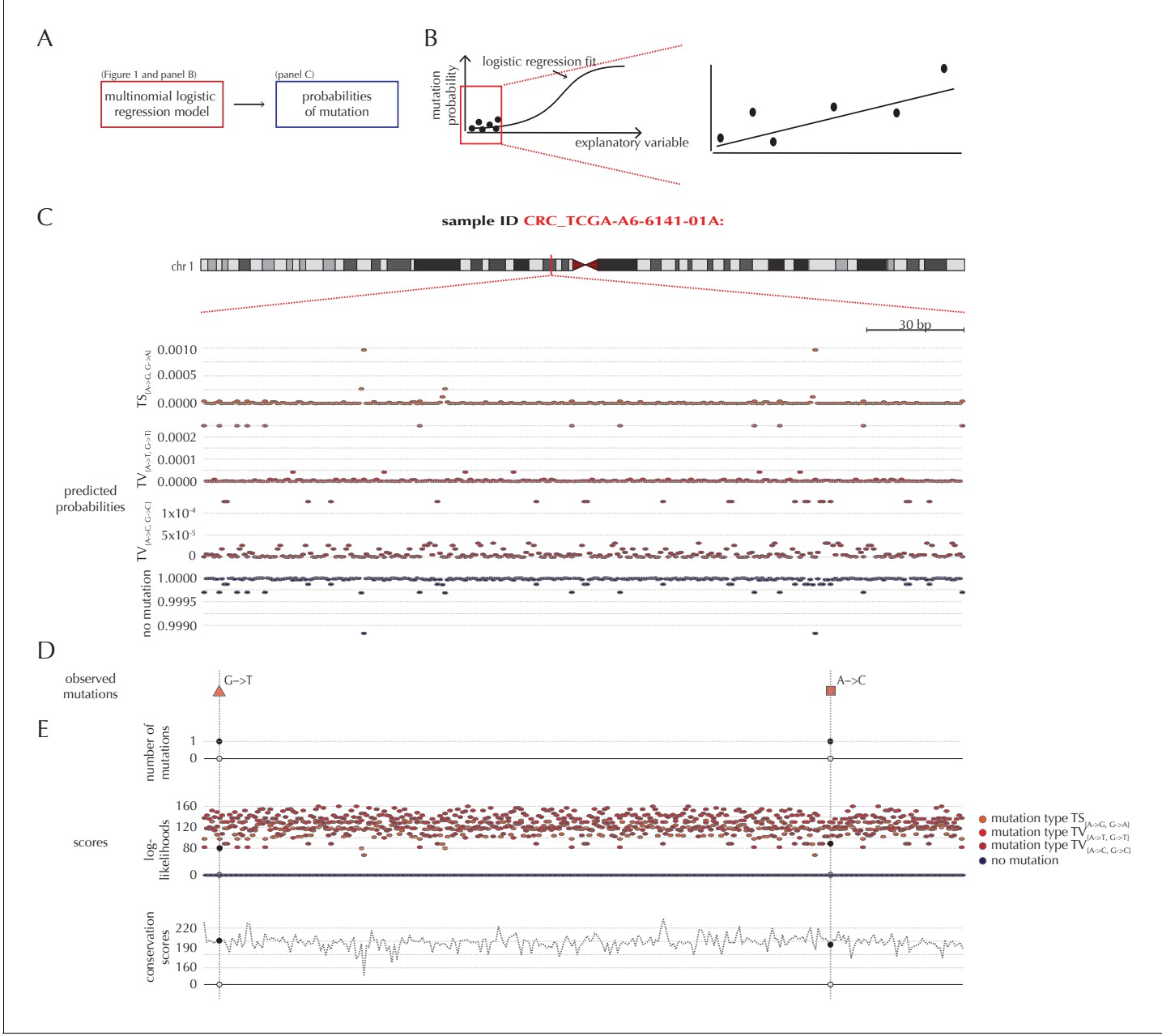

**Figure 2.** Position- and sample-specific predicted mutation rates and scoring-schemes. (**A**) A multinomial logistic regression model is used to predict the sample- and position-specific background mutation-probabilities. (**B**) The genomic annotations and the reference sequence (*Figure 1*) are used as explanatory variables in a regression fit of the somatic mutation rate. In effect, a logistic regression model is fitted for each of the four types of outcome (three types of mutation and no mutation) and combined into a multinomial logistic regression fit. Logistic regression ensures probability-predictions between zero and one. The mutation probabilities are of such small magnitude that we observe near linearity of the logistic regression curve. (**C**) Sample- and position-specific predicted mutation probabilities for each of the four outcomes in a 300 bp region of chromosome 1 (chr1:115,824,535–115,824,834) for the colorectal cancer sample CRC_TCGA-A6-6141-01A. (**D**) Observed sample-specific somatic mutations within the same region. For the sample in question, two mutations are observed; one of type $TV_{\{A\to T,\ G\to T\}}$ and one of type $TV_{\{A\to C,\ G\to C\}}$. (**E**) Sample- and position-specific scores for each of the three considered scoring schemes.

predicted position- and sample-specific mutation probabilities. This scoring scheme ensures that the more unlikely a mutational event, the higher the associated score. A third approach is to also evaluate the functional impact of mutations when defining the scores. However, for non-coding regions,

we often lack the functional understanding to interpret and predict the functional impact. We therefore illustrate this approach using phyloP, a position-specific score of evolutionary conservation (*Pollard et al., 2010*), as a proxy score for functional impact (Material and methods: Scoring schemes).

The three proposed scoring schemes assign an *observed* score value of zero (number of mutations and conservation scores), or a value close to zero (log-likelihoods), to positions with no mutations, and a positive score to positions with mutations (*Figure 2E*). The assigned scores for mutated positions depend on the mutation type and the scoring scheme. Also, for each position, all *possible* score values and the associated predicted probabilities are integrated in the calculation of the background score distribution.

## Driver detection

Although ncdDetect is designed for the analysis of non-coding elements, it can also be applied on protein-coding genes. We initially evaluate the performance of different versions of ncdDetect null models and different scoring schemes on protein-coding genes. We then use it to detect driver candidates among promoter elements, splice sites, 5' UTRs and 3' UTRs (*Table 1*, Material and methods: Candidate elements). While all the analyses presented here are pan cancer, individual cancer types can be analyzed separately.

### ncdDetect significance evaluation

With ncdDetect, significance evaluation of the observed mutations in a given genomic region of interest is performed (*Figure 3*). ncdDetect uses a two-step algorithm in which sample-specific calculations are followed by calculations across all samples in the dataset. The output is a p-value indicating if the region of interest is under recurrent positive selection across the sample set (*Figure 3A*).

The test statistic used in the significance evaluation performed by ncdDetect is the *observed score*. This value is defined as the sum of sample- and position-specific observed scores across the specific element that is being tested. For a given sample and a given position, the observed score will depend on the chosen scoring scheme. For instance, the scoring scheme using the number of mutations will always give a score of 1 to a mutated position, and a score of 0 to an unmutated position. The scoring scheme using phyloP will give a score corresponding to the position-specific phyloP value to a mutated position, and a score of 0 to an unmutated position. Thus, the observed score for a specific element will depend on the chosen scoring scheme.

The observed score is significance evaluated in the *background score distribution*. Again, the shape of this distribution will depend on the chosen scoring scheme: All possible sample- and position-specific scores for the chosen scoring scheme are combined with the sample- and position-specific mutational probabilities to form the background score-distribution.

The algorithm to obtain the background score-distribution works as follows: for a specific sample *i*, the genomic region of interest is annotated with position-specific probabilities of mutation and scores (*Figure 3B–C*). As noted above, the observed score of sample *i* is defined as the sum of observed scores across all positions in the region (*Figure 3D*). In the current implementation, only the two highest scoring mutations are considered for each sample. The position-specific mutational probabilities and scores are aggregated using mathematical convolution; a method to efficiently calculate the exact probability of observing a given sample-level score by summing up the probabilities

**Table 1.** Overview of elements analyzed with ncdDetect. Regions located on chromosome X and Y are excluded from the analyses (Material and methods: Candidate elements).

| Element type | Number of elements | Median element length (bps) | Percentage of genome covered |
|---|---|---|---|
| protein-coding genes | 20,153 | 1296 | 1.19 |
| promoter elements | 20,052 | 848 | 0.69 |
| splice sites | 18,682 | 30 | 0.03 |
| 3' UTRs | 19,346 | 1007 | 1.06 |
| 5' UTRs | 19,078 | 259 | 0.25 |

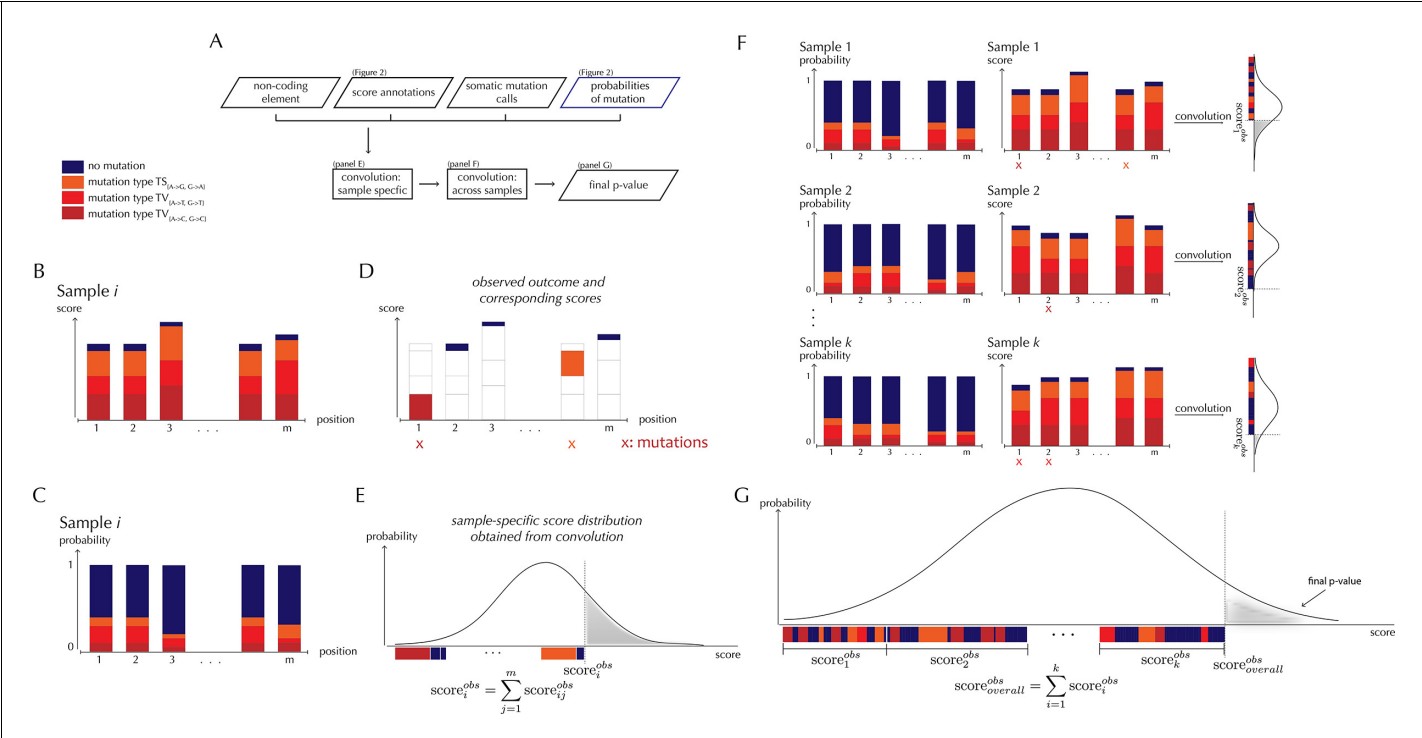

**Figure 3.** ncdDetect analysis concepts. (**A**) Flowchart of the algorithmic steps of ncdDetect. Panels B through E show the sample-specific calculations, while panels F and G show the calculations across samples. (**B**) The genomic candidate region is annotated with position- and sample-specific scores. The values of these scores depend on the choice of scoring scheme. (**C**) The region is also annotated with sample- and position-specific predicted mutation probabilities. These probabilities are predicted by the null model and does not depend on the choice of scoring scheme. (**D**) The observed score of the sample is defined as the sum of the scores associated with the observed mutational events. Scores based on number of mutations and conservation will assign non-mutated positions with a score-value of zero. Scores based on log-likelihoods will assign non-mutated positions with a positive score-value, which in practice will be near zero. (**E**) The sample-specific background score-distribution is obtained by convolution. (**F**) Sample-specific calculations are carried out for each individual sample in the dataset. (**G**) The overall background score-distribution is obtained by convolution of the individual-sample distributions. This figure is conceptual and not based on actual data. *Figure 4D–F* are real examples of background score-distributions.

The following figure supplement is available for figure 3:

**Figure supplement 1.** Illustration of time complexity of the ncdDetect algorithm.

of all possible combinations of positional outcomes that could lead to it (*Grinstead and Snell, 1997*). The use of convolution is inspired by previously published protein-coding driver detection methods (*Lawrence et al., 2013*; *Dees et al., 2012*). These calculations lead to the sample-specific background score-distribution (*Figure 3E*). By repeating this process, background score-distributions are found for each individual sample (*Figure 3F*). These distributions are aggregated, again using convolution, to yield the overall background score-distribution across samples. The individual sample-specific observed scores are summed to give the overall observed score, which is significance evaluated in the overall background score-distribution. (*Figure 3G*) (more details are given in Appendix, section 3).

## Protein-coding driver detection and model selection

To build a robust background null-model and evaluate the performance of ncdDetect, we apply it to protein-coding genes (*Figure 4*). While we lack a well-established true-positive driver set for the non-coding part of the genome, the COSMIC Cancer Gene Census (*Cosmic, 2016*) provides that for the protein-coding genes. As a performance measure, we therefore use the fraction of COSMIC genes recalled among the ncdDetect candidate sets.

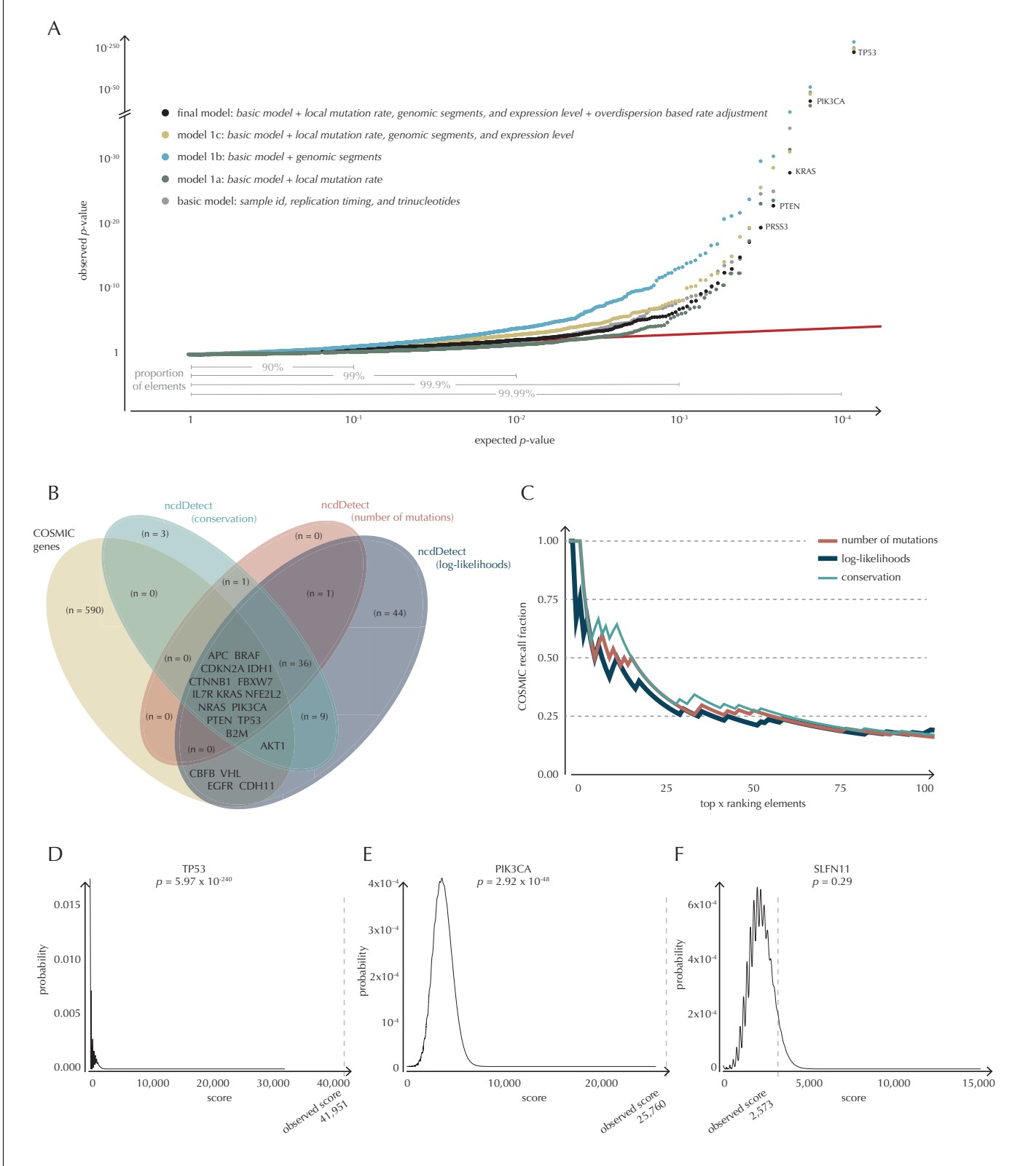

**Figure 4.** Analysis of protein-coding genes to evaluate ncdDetect performance. (A) The final null model is obtained through forward model-selection. The QQ-plot shows the p-values of all genes (n = 19,256) plotted against their uniform expectation under the null for each of the five models considered. Deviations from the expectations (red identity line) are seen for a varying proportion of the genes (0.5–10%). Results are shown for conservation scores. Similar plots for log-likelihoods and number of mutations are shown in **Figure 4—figure supplement 1**. (B) Venn diagram showing

*Figure 4 continued on next page*

*Figure 4 continued*

the overlap between protein-coding genes called as drivers by ncdDetect (q<0.10) for the three scoring schemes and the COSMIC Gene Census list. (**C**) COSMIC Gene Census recall plot. The fraction of COSMIC genes recalled in the top ncdDetect candidates. (**D–F**) The two most significant genes called by ncdDetect are *TP53* and *PIK3CA*. An example of a gene not called significant is *SLFN11*. For each of these, the convoluted background score-distributions are shown together with the observed scores and resulting p-values.

The following source data and figure supplements are available for figure 4:

**Source data 1.** P-values obtained on protein-coding genes for each of the five models considered.

**Source data 2.** COSMIC Gene Census recall data.

**Source data 3.** The background score distribution for the protein-coding gene TP53 obtained with conservation scores.

**Source data 4.** The background score distribution for the protein-coding gene PIK3CA obtained with conservation scores.

**Source data 5.** The background score distribution for the protein-coding gene SLFN11 obtained with conservation scores.

**Figure supplement 1.** Analysis of protein-coding genes to evaluate ncdDetect performance for scores defined by log-likelihoods and the number of mutations.

**Figure supplement 1—source data 1.** P-values obtained on protein-coding genes for each of the models considered.

**Figure supplement 2.** The p-values (based on conservation scores) plotted as a function of the total number of mutations across samples observed per bp for all protein-coding genes.

**Figure supplement 2—source data 1.** For each protein-coding gene, the gene length, the number of observed mutations across all 505 samples, and the p-value obtained using conservation scores are given.

---

With ncdDetect, multiple hypothesis tests are performed. For example, protein-coding driver detection requires significance evaluation of 19,256 genes. In order to evaluate all these tests simultaneously, QQ-plots are used to assess the distribution of the p-values and the number of true hypotheses (*Schweder and Spjotvoll, 1982*). In these plots, the observed p-values are plotted against the expected (uniform) p-values of the null distribution. P-values, which follow the expected uniform distribution, will thus fall on the identity line, while smaller p-values will deviate from this line. Per construction, 90% of the expected values lie in the interval $[1, 10^{-1}]$, 99% lie in the interval $[1, 10^{-2}]$, etc. (*Figure 4A*).

## Model selection

The final model underlying ncdDetect is determined through a forward model-selection procedure. In each step, position- and sample-specific probabilities are predicted for the protein-coding genes, which are then evaluated with ncdDetect (*Figure 4A*, *Figure 4—figure supplement 1*). The *basic model* includes the genomic annotations sample id, replication timing and trinucleotides as these are all known to correlate with mutation rate. The resulting p-values appeared slightly inflated. To increase robustness of the predicted mutation-probabilities, we defined *model 1a* by adding the variable local mutation rate to the basic model. This addition resulted in less inflated p-values. However, the p-values were below the identity line in the QQ-plot for more than 99% of the analyzed genes, indicating that the predicted probabilities of mutation were too large. As we found the somatic mutation rate is elevated in intergenic regions compared to other functional elements, we defined *model 1b* by adding the variable genomic segment to the basic model (*Figure 1—figure supplement 1*). This had the desired effect of decreasing the final p-values, although leading to severe inflation. We defined *model 1c* that extended the *basic model* with the local mutation rate, genomic segment, and tissue-specific gene expression level. This lowered the p-values, although a small amount of inflation was still observed.

Since we do not know all the relevant genomic annotations that correlate with mutation rate for all of our samples, it is unavoidable that we observe a difference between the actual and predicted

mutation rate (*Appendix 1—figure 1*). The effect of this difference will be accumulated along elements, and even small biases in the predicted versus observed mutation rate may become significant if elements are sufficiently long (*Appendix 1—figure 2*). The difference between the predicted and observed mutation rate will cause overdispersion of the mutation rate. In the *final model*, we thus correct for overdispersion by adjusting the sample- and position-specific probabilities of mutation with an overdispersion-based rate adjustment (Materials and methods: An overdispersion-based mutation rate adjustment, Appendix section 1). The resulting p-values follow the expected uniform distribution, with less extreme p-values for the top-ranked genes than for the previous models.

## Recall of known protein-coding drivers

The p-values obtained with ncdDetect are corrected for multiple testing using a false discovery rate of 10% (*Benjamini and Hochberg, 1995*). The resulting ncdDetect candidate protein-coding drivers are compared to the COSMIC Gene Census list for each of the three proposed scoring schemes. We call 64 protein-coding genes significant using the conservation scores of which 15 (23%) are in COSMIC. In contrast, we call 109 protein-genes significant with log-likelihoods of which 19 (17%) are in COSMIC, and 52 with the number of mutations of which 14 (27%) are in COSMIC (*Figure 4B*, *Supplementary files 1–3*). The mean number of mutations per bp is on average eight times higher for the COSMIC genes detected by ncdDetect compared to the undetected COSMIC genes (*Figure 4—figure supplement 2*). The three proposed scoring schemes have similar recall graphs, although the use of conservation scores appears the most sensitive as it generally recalls the highest fraction of COSMIC genes (*Figure 4C*). For example, in the top-15 protein-coding genes called by ncdDetect with conservation scores, nine are COSMIC genes. This number is seven for the number of mutations, and seven for log-likelihoods (*Figure 4C*, *Figure 5—figure supplement 1A*). The use of log-likelihoods results in the highest number of elements called significant across most element types (*Figure 5—figure supplement 2*).

As the mutational process is stochastic, it varies which drivers are involved in cancer development, both within and between cancer types. COSMIC genes are identified from analyses across many individual cancer types and a large fraction are likely not drivers in the particular set of cancer samples analyzed here. Furthermore, there might exist true protein-coding cancer drivers not yet included in COSMIC. Out of the three proposed scoring schemes, the conservation scores appear to have the highest sensitivity. It is more conservative than the log-likelihoods, as it finds fewer significant protein-coding genes. Furthermore, it is compelling that the use of this scoring scheme incorporates a measure of functional mutational impact in the driver significance evaluation. In light of these considerations, we focus on the results obtained with conservation scores in the following, and include the remaining two scoring schemes for comparison.

To give an impression of how the calculated background score-distributions behave in practice, we highlight a few examples (*Figure 4D–F*). The top-two protein-coding genes called by ncdDetect are *TP53* (spanning 1378 bps) and *PIK3CA* (spanning 3207 bps), which are both well-known cancer driver genes. An example of a protein-coding gene not called significant by ncdDetect is *SLFN11* (spanning 2706 bps). The smoothness of the overall score-distribution is related to the length of the gene.

The performance of ncdDetect on protein-coding genes was benchmarked against a recent noncoding cancer driver detection method, ExInAtor (*Lanzós et al., 2017*) (Appendix, section 2). In general, ExInAtor predicts much fewer candidates than ncdDetect, and thus has a lower false-positive rate. However, ncdDetect performs better at ranking genes compared to ExInAtor (*Appendix 1—figure 3*).

## Non-coding driver detection

Although the functional impact of non-coding mutations in cancer is not yet fully understood, it is widely believed that they may play an important role in cancer development (*Diederichs et al., 2016*). Here, we apply ncdDetect to gene-associated non-coding elements of various types (promoter elements, splice sites, 3' UTRs and 5' UTRs) to evaluate their cancer driver potential (*Figure 5*, *Figure 5—figure supplement 1*, *Supplementary files 1–3*, Material and methods: Candidate elements).

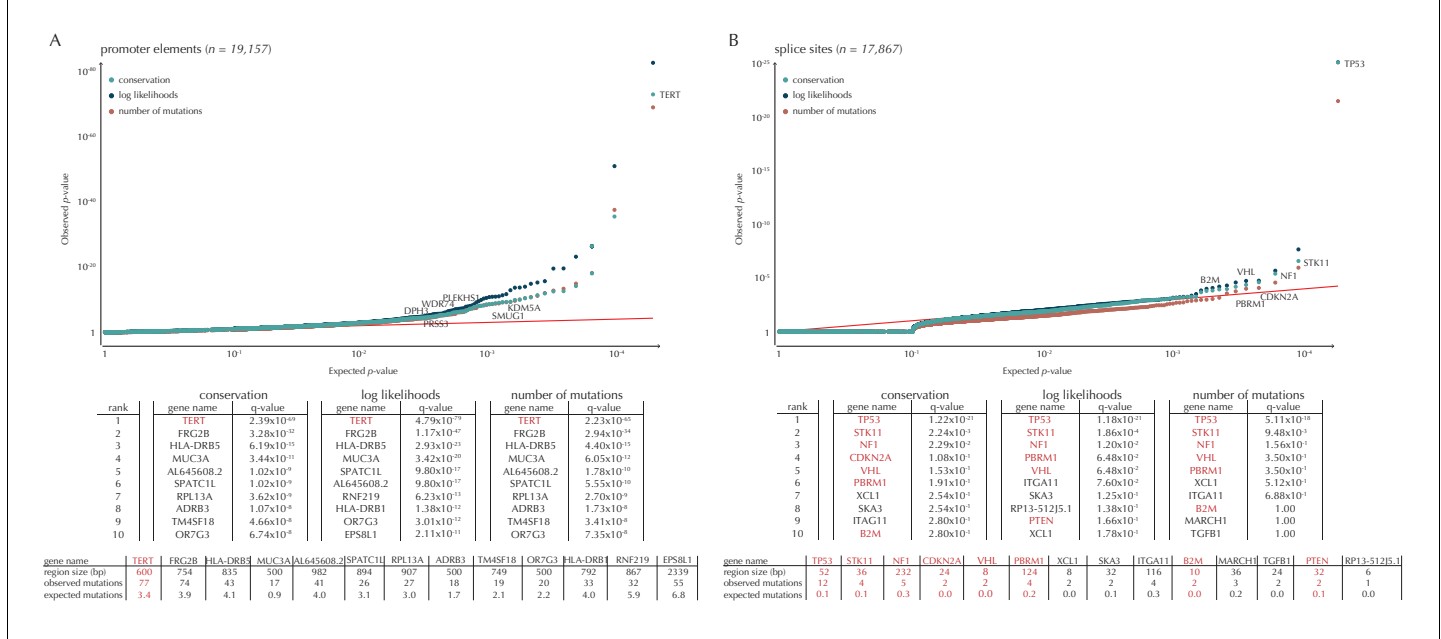

**Figure 5.** Q-values and top-ten ranking non-coding elements for each of the three proposed scoring schemes. The results discussed in the text relate to conservation scores. Non-coding elements associated to COSMIC genes are highlighted in red. For each element, the region size is given together with the observed number of mutations and the expected number of mutations under the null model. (**A**) The QQ-plot shows the p-values for all promoter elements (n = 19,157) plotted against their uniform expectation under the null. One hundred and sixty promoter elements are found to be significant. (**B**) QQ-plot of p-values for all splice sites (n = 17,867). The p-values do not follow the expectation under the null. This is explained by the fact that 90% of all splice sites carry no mutations. Three splice sites come up significant with ncdDetect after correcting for multiple testing.

The following source data and figure supplements are available for figure 5:

**Source data 1.** P-values obtained on promoters and splice sites using conservation scores.

**Figure supplement 1.** Q-values and top-ten ranking elements for each of the three proposed scoring schemes.

**Figure supplement 1—source data 1.** P-values obtained on protein-coding genes, 3' UTRs and 5' UTRs using conservation scores.

**Figure supplement 2.** The number of elements called significant for each of the three proposed scoring schemes, for each of the defined element types.

**Figure supplement 2—source data 1.** The number of elements called significant for each of the three proposed scoring schemes, for each of the defined element types.

**Figure supplement 3.** Length distributions of all defined element types.

**Figure supplement 3—source data 1.** The length of each of the analyzed elements.

## Recall and function of previously described non-coding drivers

Promoter mutations might dysregulate gene expression in cancer. In particular, such mutations might affect the expression levels of tumor suppressor genes or oncogenes (*Diederichs et al., 2016*). The average mutation rate in the analyzed promoter elements is 7.0 mutations per mega base (Mb) per sample (*Figure 1—figure supplement 1*). Of the investigated non-coding element types, the promoter elements have the most significant calls in the ncdDetect analyses. Approximately 1% of the evaluated promoter elements have a more significant p-value than expected under the null (*Figure 5A*). We find a total of 160 significant (q < 0.10) promoter elements. Within these, the observed mutation rate is 31.3 mutations per Mb per sample, which is a fourfold increase of the mutation rate among all promoter elements. Of the promoter elements, the *TERT* promoter is

ranked most significant (q = $2.4 \times 10^{-69}$, ncdDetect). The promoter of the *TERT* gene is known to play an important role in telomerase expression, and cancers with *TERT* promoter mutations have been shown to exhibit an elevated expression of the *TERT* gene. This increased expression might ensure telomere maintenance, believed to enable cancer cells to divide (*Heidenreich et al., 2014*). Two other identified promoter elements are *WDR74* (q = $4.1 \times 10^{-4}$, ncdDetect) and *PLEKHS1* (q = $4.3 \times 10^{-5}$, ncdDetect). Mutations in the promoter region of *WDR74* have been associated with increased gene expression and are thought to have functional relevance for tumorigenesis (*Khurana et al., 2013*). Mutations in the *PLEKHS1* promoter are also previously found to be significant in non-coding driver screens (*Weinhold et al., 2014*; *Melton et al., 2015*). We note that out of the 863 whole genomes analyzed in *Weinhold et al. (2014)*, 356 are sequenced by TCGA. These samples appear to be a subset of the 505 samples analyzed in the present work, and the data sets are thus not completely independent. In total, 29 of the 160 significant promoter elements called with ncdDetect are previously found to be significant in non-coding cancer driver studies (*Weinhold et al., 2014*) (*Appendix 1—figure 4*). As a benchmark of the performance of ncdDetect on regulatory non-coding regions, we compared our results on promoter elements to those obtained with another non-coding cancer driver detection method, LARVA (*Lochovsky et al., 2015*). The ncdDetect promoter candidates that are not detected by LARVA include the previously described *WDR74*, *PLEKHS1* and promoters of COSMIC genes (Appendix, section 2).

Another class of non-coding mutations are splice site mutations. They might disrupt the splicing code and have been linked to cancer development (*Srebrow and Kornblihtt, 2006*). The destruction of a splice site will typically introduce stop codons or frameshifts and ruin the function of the translated protein. The splice site mutation rate is 5.1 mutations per Mb per sample. Three splice sites are found significant in this analysis (*Figure 5B*). As many as 90% of the splice sites have zero observed mutations across the 505 samples. By construction, this means that the resulting p-values are 1, and the p-value distribution is thus not uniform. More samples would increase the detection power in these cases. Interestingly, in the top-ten ranking splice sites, we see a highly significant enrichment of splice sites associated to COSMIC genes (p=$6.1 \times 10^{-9}$, Fisher's exact test). Within the three significant splice site elements, the mutation rate is 130.0 mutations per Mb per sample, corresponding to a 25-fold increase of the mutation rate among all splice site elements. Splicing mutations in *TP53* are previously described in cancer studies (*Lee et al., 2010*; *Varley et al., 2001*). Here, we observe that 12 samples from six different cancer types are mutated in the 52 bps that make up the splice sites of *TP53*. These splicing mutations are highly significant (q = $1.2 \times 10^{-21}$, ncdDetect), and might lead to inactivation of the tumor suppressor *TP53* gene (*Eicheler et al., 2002*).

Finally, we investigate somatic mutations in the 3' and 5' UTRs (*Figure 5—figure supplement 1B–C*), which regulate mRNA stability and translation. UTR mutations might disrupt binding sites for miRNAs and RNA-binding proteins and thereby affect post-transcriptional regulation. They might also alter the structural conformations of the UTRs, which have previously been associated with cancer (*Diederichs et al., 2016*). The average number of mutations is 6.4 per Mb per sample for 5' UTRs and 7.1 per Mb per sample for 3' UTRs. We find a total of 16 significant 3' UTRs and 86 significant 5' UTRs (*Figure 5—figure supplement 1B–C*). Within the significant 5' UTRs, the mutation rate is 36.3 mutations per Mb per sample, a sixfold increase compared to all 5' UTRs. For the significant 3' UTRs, the mutation rate is 20.6 mutations per Mb per sample, which is a threefold increase of the average 3' UTR mutation rate. Two of the called 3' UTRs (*DRD5* and *PCMTD1*) have previously been detected in cancer driver studies (*Weinhold et al., 2014*). This is also the case for 12 of the 86 called 5' UTRs, one of them *SDHD* (q = $2.3 \times 10^{-3}$, ncdDetect) (*Appendix 1—figure 4*). A recent study identified the promoter region as well as the 5' UTR of *SDHD* to be potential cancer drivers in melanoma. In particular, promoter mutations of *SDHD* were shown to be associated with reduced gene expression and poor survival prognosis (*Weinhold et al., 2014*). In the present data set, we observe six mutated melanoma samples in the 5' UTR of *SDHD*, which covers 135 bps.

## Case studies

The absence of a true-positive driver set for the non-coding part of the genome means that we must find alternative ways to validate the driver potential of the candidates found by ncdDetect. We thus

seek to support the significance of the candidate elements and further characterize them with evidence from two additional data sources.

A first approach is to analyze the effects of mutations on gene expression. To be able to look up individual driver candidates, we gather expression values for each of the 505 considered whole genome samples (*Fredriksson et al., 2014*) and perform a Wilcoxon rank sum test for top-ranking candidates of each element type. To further support the findings, we obtain mutation calls and expression values from the larger set of TCGA exomes (*Weinstein et al., 2013*; *Supplementary file 7*) and likewise perform a rank sum test on these data (*Supplementary file 5*, Material and methods: Expression analysis). Reassuringly, we recover known differences in *TERT* gene expression levels between samples mutated and not mutated in the promoter region of the gene for the 505 whole genome samples (q = $1.4 \times 10^{-3}$, Fisher's method) (*Vinagre et al., 2013*). Similarly for the TCGA exome samples, splice-site mutations in *TP53*, which are known to drive cancer (*Varley et al., 2001*; *Lee et al., 2010*), correlated with differences in gene expression levels (q = $2.3 \times 10^{-2}$, Fisher's method).

Another approach we take to validate the ncdDetect candidates is to look at correlation between mutation status and survival data for both the 505 whole genome samples and the TCGA exomes. For this purpose, we download clinical data from the TCGA data portal (*TCGA Data Portal, 2016*; *Supplementary file 7*). For a particular candidate driver, we test the significance of the difference in survival between mutated and non-mutated samples using a one-sided Log-rank test on the Kaplan-Meier estimated survival curves (*Supplementary file 6*, Material and methods: Survival analysis). This recovers some known prognostic markers, such as *TP53* where splice site mutations correlate with a significant decrease in survival (q = $1.0 \times 10^{-1}$, Fisher's method) (*Yang et al., 2013*). In the analysis of the 505 whole genome samples, we furthermore observe a significant decrease in survival associated with *HLA-DRB1* promoter mutations (q = $2.0 \times 10^{-2}$, Fisher's method). Although this finding is potentially interesting, further investigation of this candidate is beyond the scope of this paper, as genotyping in HLA regions is challenging due to the highly polymorphic nature of these genes (*Ehrenberg et al., 2014*).

In the following, we study a number of the top-ranking non-coding ncdDetect driver candidates in detail. For each of them, we further evaluate their driver potential by including results from expression analysis and survival analysis.

## SMUG1 mutations and a uracil-DNA glycosylase deficiency mutational signature

We observe 19 mutations in the 997 bp-long *SMUG1* promoter-region, which is approximately seven times more than expected under the null model (q = $1.1 \times 10^{-6}$, ncdDetect) (*Figure 6A*). The mutations are distributed among 14 samples from three different cancer types (one breast cancer sample, two colorectal cancer samples and eleven melanoma samples). As 15 out of 16 of the melanoma mutations are of type C→T in a CC context (or its reverse complement), they are consistent with the mutational signature of ultraviolet (UV) light (*Alexandrov and Stratton, 2014*). They may be a result of a mutational mechanism (*Sabarinathan et al., 2016*); however, they may also affect *SMUG1* function.

*SMUG1* is involved in base excision repair (BER). Together with *UNG*, it acts in BER as an uracil-DNA glycosylase, that is, an enzyme that removes uracil from DNA (*Visnes et al., 2009*). Uracil in DNA arises from spontaneous deamination of non-methylated cytosine, which causes the occurrence of U:G mismatches. If unrepaired, they give rise to G→A transition mutations. Mouse cell line experiments have shown additive effects of *SMUG1* and *UNG* inactivation on G→A mutation rates (*An et al., 2005*). Furthermore, *UNG* and *SMUG1* expression has recently been found to correlate negatively with genomic uracil levels in B cell lymphomas (*Pettersen et al., 2015*). We therefore hypothesize that *SMUG1* mutations may affect the rate of G→A (and C→T) mutations. To investigate this further, we define the *uracil-DNA glycosylase deficiency signature* as the proportion of G→A (including C→T) mutations outside CpG sites (*Figure 6B*). For melanoma samples, we further deduct C→T mutations in a CC context, as potentially induced by UV light (Material and methods: Enrichment of G→A mutations in *SMUG1* mutated samples).

There is a tendency for an increased value of the uracil-DNA glycosylase deficiency signature statistic in *SMUG1* mutated melanoma samples (p=$8.2 \times 10^{-2}$, one-sided Wilcoxon rank sum test), and a significantly increased value of this statistic for the one *SMUG1* mutated uterus cancer sample

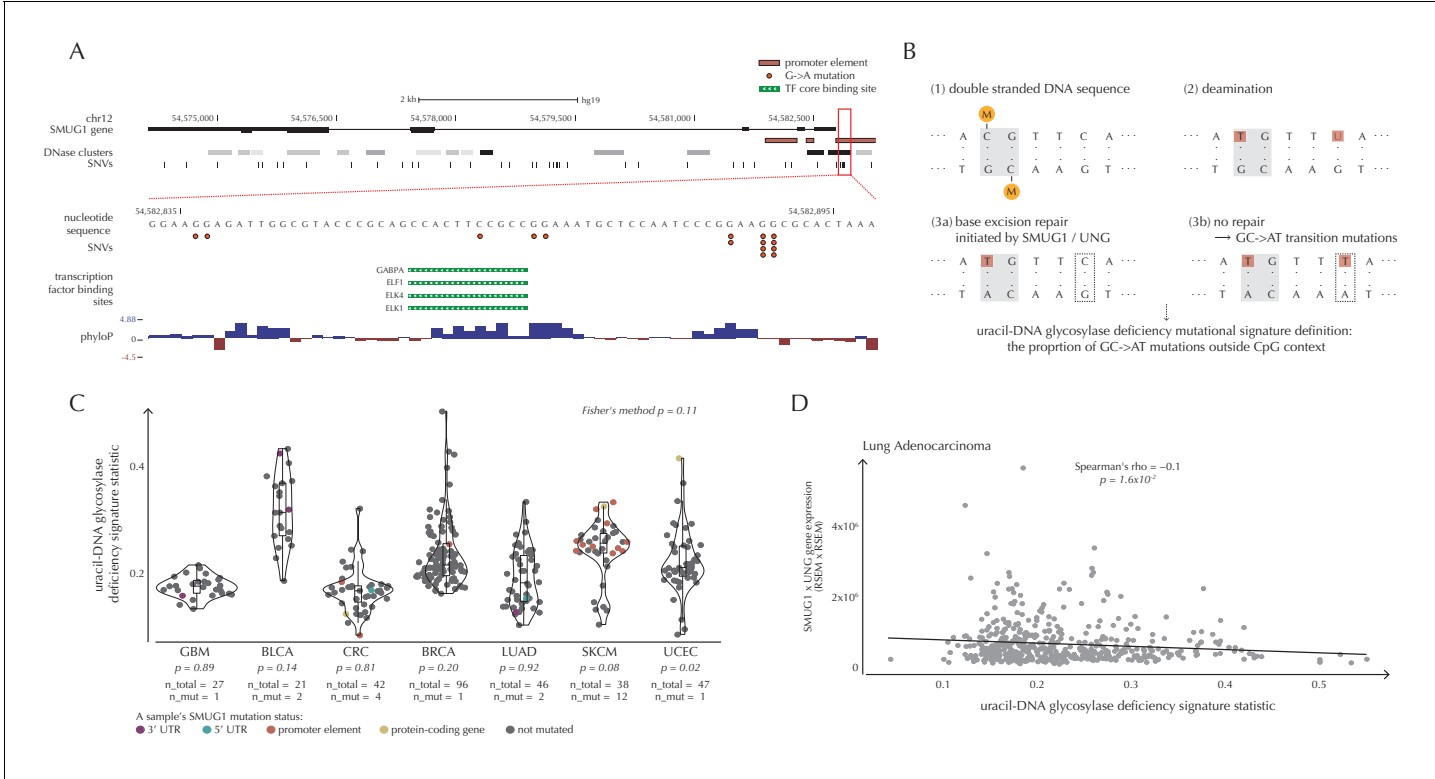

**Figure 6.** *SMUG1* mutations and base excision repair. (**A**) Genomic overview of *SMUG1* showing its promoter region (*Kent et al., 2002*). The DNase clusters track shows DNase hypersensitive regions where the darkness is proportional to the maximum signal strength observed in any cell line (*ENCODE Project Consortium, 2012*). The transcription-factor-binding sites (TFBSs) track shows core regions of transcription factor binding (*Gerstein et al., 2012*). The phyloP track shows evolutionary conservation of positions (*Pollard et al., 2010*). (**B**) Uracil-DNA glycosylase deficiency signature definition: (1) Cytosines may be methylated (orange circles) at CpG sites (gray box). (2) Spontaneous deamination (red boxes) of non-methylated cytosine results in uracil, causing U:G mismatches. Spontaneous deamination of methylated cytosine results in thymine, causing T:G mismatches. (3a) *SMUG1* and *UNG* are uracil-DNA glycosylases, which, via base excision repair, will repair the U:G mismatches caused by deamination. (3b) If unrepaired, the U:G mismatches will result in G→A mutations. (**C**) A one-sided Wilcoxon rank sum test is performed per cancer type to investigate if samples with a *SMUG1* mutation have a higher value of the uracil-DNA glycosylase deficiency signature statistic than samples without such a mutation. The analysis is based on the 505 whole genome TCGA samples. Each dot represents a sample, and the color represents the *SMUG1*-associated mutated element. (**D**) Correlation between the uracil-DNA glycosylase deficiency signature statistic and the product of *SMUG1* and *UNG* gene expression using TCGA exome data for lung adenocarcinoma.

The following source data and figure supplement are available for figure 6:

**Source data 1.** The defined uracil-DNA glycosylase deficiency signature statistic for each sample of the cancer types GBM, BLCA, CRC, BRCA, LUAD, SKCM and UCEC.

**Source data 2.** The defined uracil-DNA glycosylase deficiency signature statistic, as well as SMUG1 gene expression, UNG gene expression, and SMUG1xUNG gene expression for TCGA exome samples.

**Figure supplement 1.** Examples of correlation between the uracil-DNA glycosylase deficiency signature statistic and *SMUG1* gene expression (first column), *UNG* gene expression (second column) and the product of *SMUG1* and *UNG* gene expression (third column) using TCGA exome data for seven different cancer types (rows).

(p=2.1 × 10$^{-2}$, one-sided Wilcoxon rank sum test), compared to samples not harbouring a *SMUG1* mutation (*Figure 6C*). This is also the case when restricting the analysis to only include coding and splice site mutations (melanoma: p=5.3 × 10$^{-2}$, uterus cancer: p=2.1 × 10$^{-2}$, one-sided Wilcoxon rank sum tests). These findings indicate that *SMUG1* mutations might perturb the uracil-DNA glycosylase function.

We further hypothesize that *SMUG1* and *UNG* expression may correlate with the uracil-DNA gly-cosylase deficiency signature statistic. With the expression data available for the 505 analyzed TCGA samples, we are unable to detect a significant correlation between gene expression of *SMUG1* or *UNG* and the signature statistic (*SMUG1*: p=9.4 × $10^{-1}$, *UNG*: p=1.0 × $10^{-1}$, Fisher's method). To further investigate expression correlations, we look at the larger data set of TCGA exomes. From these data, the uracil-DNA glycosylase deficiency signature statistic is seen to be negatively corre-lated with *SMUG1* gene expression (p=3.8 × $10^{-2}$, Fisher's method), and with *UNG* gene expression (p=5.1 × $10^{-4}$, Fisher's method). As *SMUG1* and *UNG* are thought to have complementary roles in BER (*Pettersen et al., 2007*), we also investigate the correlation between the signature statistic and the product of *SMUG1* and *UNG* gene expression, which is negative and also significant (p=2.4 × $10^{-3}$, Fisher's method) (*Figure 6D*, *Figure 6—figure supplement 1*).

Finally, we investigate whether survival correlates with *SMUG1* mutation status. With the present data set, we are not able to detect such a pattern (p=8.2 × $10^{-1}$, Fisher's method).

The observed correlations combined with the existing literature (*Pettersen et al., 2015*; *An et al., 2005*) suggest that mutations that functionally impact *SMUG1* and *UNG* may cause a mutational phenotype as captured by the defined deficiency signature. However, further validation must await availability of larger sets of cancer genomes.

## Promoter and UTR candidates where mutations associate with decreased survival

The 5' UTR of *CD1A* spans 533 bp. In the region we observe 11 mutations from 10 different sam-ples, distributed across eight different cancer types. This corresponds to five times the amount of mutations expected under the null model (q = 1.1 × $10^{-2}$, ncdDetect). For TCGA exome melanoma samples, we observe a highly significant decrease in survival associated with mutations in the region (p=1.2 × $10^{-9}$, Log-rank test) (*Figure 7A*), which is top-ranked among the non-coding regions tested (*Supplementary file 6*). CD1 proteins present antigens to T cells and are involved in eliciting adap-tive immune responses. They are distantly related to HLA (MHC) proteins and similarly bind T cell receptors; however, they display glycoproteins and small molecules instead of peptides (*Van Rhijn et al., 2015*). Intriguingly, *CD1A* is generally lowly expressed in healthy tissue with high expression particularly in skin (*GTEx Consortium, 2013*), where it is found in the antigen-presenting Langerhans cells. *CD1A* has previously been implicated with cancer development, with expression and positive correlation to survival reported for some cancer types (*Coventry and Heinzel, 2004*). Although we cannot functionally interpret the observed TCGA melanoma mutations, the strong association with survival suggests potential clinical relevance, not-the-least given the success of immunotherapy in melanoma (*Drake et al., 2014*).

A total of 22 mutations are observed in the 1976 bp-long promoter region of *PRSS3*. This is four times more mutations than expected under the null model, and they occur in 13 samples from seven different cancer types (q = 1.1 × $10^{-2}$, ncdDetect). Previous studies have established the role of *PRSS3* in the progression of pancreatic and ovarian cancer (*Jiang et al., 2010*; *Ma et al., 2015*). Not only the promoter region of this gene comes out significant in the driver detection screen; this is also the case for its 3' UTR (q = 1.4 × $10^{-5}$, ncdDetect) as well as its protein-coding gene (q = 6.8 × $10^{-20}$, ncdDetect). Based on the TCGA exome set, we observe a significantly decreased survival for head and neck cancer (HNSC) samples mutated in the promoter region of *PRSS3* (p=1.8 × $10^{-3}$, Log-rank test) (*Figure 7B*) as well as in the *PRSS3* coding gene (p=1.2 × $10^{-2}$, Log-rank test). We also observe a tendency for decreased survival among melanoma samples with 3' UTR mutations (p=8.3 × $10^{-2}$, Log-rank test).

The 3' UTR of *SEC14L1* spans 3052 bps and contains 31 mutations from 27 samples distributed across 10 different cancer types. This is approximately four times the amount of expected mutations (q = 8.9 × $10^{-5}$, ncdDetect), and the majority (58%) of these are found in breast- and colorectal can-cer. Although little is known about *SEC14L1* in cancer, one study hypothesized that altered expres-sion of the gene could contribute to breast tumorigenesis (*Kalikin et al., 2001*). Another study found *SEC14L1* to be overexpressed in prostate cancer (*Burdelski et al., 2015*). For TCGA exome HNSC samples, we find a significant decrease in survival associated with mutations in the 3' UTR region of the gene (p=5.8 × $10^{-5}$, Log-rank test).

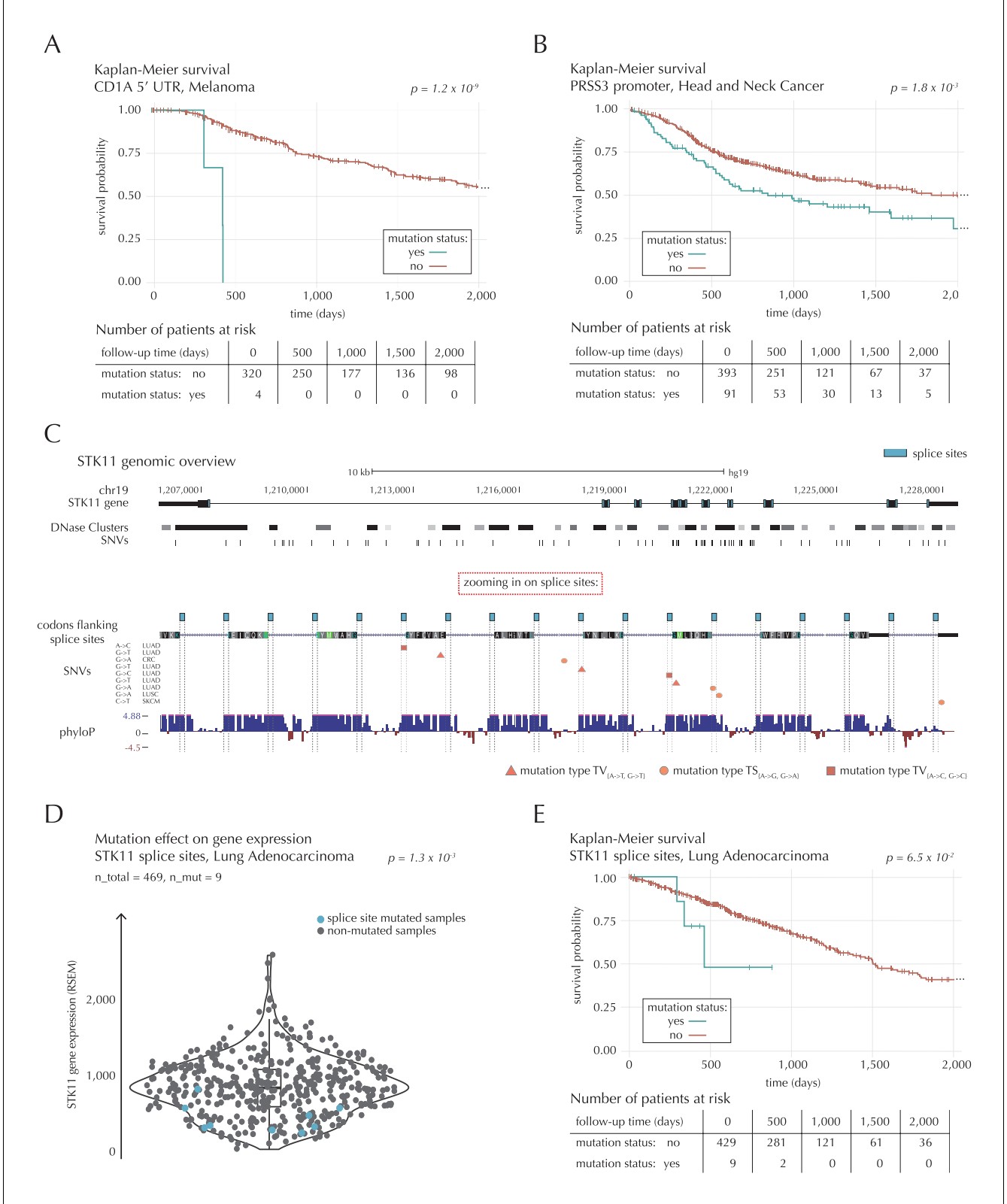

**Figure 7.** Survival- and expression analysis of *CD1A*, *PRSS3* and *STK11* mutations. (**A**) Kaplan-Meier survival curves for melanoma samples with and without mutations in the 5' UTR of *CD1A*. For illustration purposes, the data are shown for a follow-up time of 2000 days, at which point 98 out of 324 patients (30%) are still at risk. The analysis is based on the TCGA exome sample set. (**B**) Kaplan-Meier survival curves for HNSC patients with and without *PRSS3* promoter mutations. The data are shown for a follow-up time of 2000 days, at which point 42 out of 484 patients (9%) are still at risk. The

*Figure 7 continued on next page*

*Figure 7 continued*

analysis is based on the TCGA exome sample set. (**C**) Genomic overview of *STK11*, zooming in on its combined splice sites region. The phyloP track shows evolutionary conservation of positions. (**D**) A two-sided Wilcoxon rank sum test is performed for LUAD samples from the TCGA exome sample set, to investigate if samples mutated in the splice site region of *STK11* have a different gene expression level than samples without such mutations. (**E**) Kaplan-Meier survival curves for LUAD samples with and without *STK11* splice site mutations. The data are shown for a follow-up time of 2000 days, at which point 36 out of 438 patients (8%) are still at risk. The analysis is based on the TCGA exome sample set.

The following source data is available for figure 7:

**Source data 1.** STK11 mutation status and STK11 gene expression (RSEM) for 469 LUAD TCGA exome samples.

## STK11 splice sites mutations and their expression correlation

The combined splice site region of *STK11* covers 36 bps and is mutated in four lung adenocarcinoma (LUAD) cancer samples, which is approximately 56 times more mutations than expected under the null model (*Figure 7C*). ncdDetect ranks the splice site region of *STK11* second among all splice sites (q = $2.2 \times 10^{-3}$, ncdDetect). *STK11* is a known COSMIC tumour suppressor gene, which has been shown to be involved in lung and cervical cancers (*Gill et al., 2011*), and very recently, splice site mutations of the gene were described in relation to cancer (*Mularoni et al., 2016*; *Wei et al., 2016*). From the 505 whole genomes analyzed here, we are unable to associate the splice site mutations of *STK11* with a changed level of gene expression (p=$4.4 \times 10^{-1}$, Fisher's method). Looking at the larger set of TCGA exomes, however, we detect a significantly lower expression level for LUAD samples mutated in the splice site region of *STK11*, compared to LUAD samples without such mutations (p=$1.3 \times 10^{-3}$, two-sided Wilcoxon rank sum test) (*Figure 7D*). We further observe a marginally significant decrease in survival associated with *STK11* splice site mutations for LUAD TCGA exome samples (p=$6.5 \times 10^{-2}$, Log-rank test) (*Figure 7E*).

## Discussion

Non-coding somatic mutations play part in tumour initiation and progression. With the advent of whole genome sequencing, the systematic screening of such mutations is possible. We have developed the method ncdDetect with the goal of detecting non-coding cancer driver elements and thereby gain an understanding of the underlying mechanisms of tumorigenesis. With ncdDetect, we model the heterogeneous neutral background mutation-rate, taking genomic annotations known to correlate with the mutation rate into account. We consider the mutational burden and functional impact to reveal signs of recurrent positive selection across cancer genomes.

The position- and sample-specific approach behind ncdDetect sets the stage for a number of distinct types of analyses. The analysis of one contiguous region is a straight-forward application of ncdDetect, as is the combined analysis of disjoint regions, potentially with vastly different background mutation-rates. The flexible setup conveniently ensures that no constraints are necessary when defining the size and location of a particular region of interest. Furthermore, the method can be used to evaluate more complex functional hypotheses than those presented here. For instance, different sets of regions in different samples can be jointly evaluated, and sample- or tissue-specific scoring schemes can be applied directly.

Not all the significant non-coding elements can be regarded as true cancer drivers. ncdDetect might falsely identify driver elements ('false positives') for both technical and biological reasons. The false positives stemming from predicting too low a mutation rate in certain regions can be reduced by adding relevant genomic annotations as explanatory variables to the null model. In general, failing to include an explanatory variable that explains variation in the mutation rate will cause too little variation in the predicted mutational probabilities. We handle such overdispersion of the mutation rate by adjusting each sample- and position-specific probability of mutation with an overdispersion-based correction factor. This improves the model fit, although some inflation appears to remain for long elements. We acknowledge that the false-positive rate among long genes is not properly controlled and likely higher than the applied FDR threshold. We therefore continue to strive to improve our model of the site-specific mutational process, which will also improve power. The observed mutation rate also varies for technical reasons. For instance, the power to call mutations and the rate

with which mutations are missed, will vary with genomic complexity, including repeats, pseudo-genes, etc. The predicted mutational probabilities can thus be further improved by including genomic annotations that correlate with the rate of either false-negative or false-positive mutation calls as explanatory variables.

Likewise, ncdDetect might miss true driver elements ('false negatives'). Especially, with the size of the current whole cancer-genome data sets, we lack statistical power to detect infrequently mutated driver elements, or driver elements that may operate within an individual cancer type. This issue will be remedied as larger sets of sequenced whole genomes become available. In the near future, more than 2500 whole genomes will be available from the Pan-Cancer Analysis of Whole Genomes (PCAWG) project. However, it is becoming evident that some instances of detected potential driver regions may be explained by local mutational mechanisms rather than recurrent selection (*Sabarinathan et al., 2016*). This emphasizes the importance of critical scrutinization and eventually independent validation of driver candidates emerging from ncdDetect.

With ncdDetect, we screen for non-coding cancer drivers and highlight cases of special interest. To gain further evidence for the identified candidates, we correlate the presence of non-coding mutations with gene expression as well as patient survival: We find that mutations in the promoter and in the coding region of a gene in the Base Excision Repair pathway, *SMUG1*, correlate with an increase of C→T mutations. We hypothesize that *SMUG1* mutations might perturb uracil-DNA glycosylase function and cause a specific mutational phenotype. Although our study is limited to correlative observations between the expected mutational signature for uracil-DNA glycosylase deficiency and mutational presence as well as gene expression, perturbation experiments in cell lines support the hypothesis (*An et al., 2005*). We also identify non-coding regulatory regions that associate with patient survival, including the potential clinically important 5' UTR of *CD1A*, the promoter and 3'UTR of *PRSS3*, and the 3'UTR of *SEC14L1*. Finally, we identify lung cancer mutations in the splice sites of *STK11* as potential driver events. By extending the analysis to the larger TCGA data set, we show that these correlate significantly with expression. The patients also show a strong tendency for poorer survival.

In this work, we have addressed the challenges associated with distinguishing driver and passenger non-coding mutations. We evaluated three different scoring schemes and found that a conservation-based scheme performed better than mutation counts and log-likelihoods in our setting. For selected candidate cases, we found a significant effect on expression levels and a significant decrease in survival for mutated samples. The combined analyses of mutational impact on expression and survival across cancer types allowed us aggregate evidence and gain power. The screen identified candidates of potential clinical relevance. However, sample sizes remain small and further studies in large independent cohorts are necessary to establish their potential as prognostic biomarkers or therapeutic targets.

As we continue to gain larger cancer genomics data sets for driver screens, accurate modelling of the mutational heterogeneity will become increasingly important. This will help control the false-positive rate as the power of the data increases. As our understanding of the general differences in mutational mechanisms between cancer types improves further, this knowledge should be incorporated in ncdDetect.

## Materials and methods

### Statistical null model

The statistical null model that enables us to predict position- and sample-specific probabilities of mutation is a multinomial logistic regression model (*Agresti, 2013*). The model is described in detail in *Bertl et al., 2017*. Logistic regression has been used to model the background mutation rate in cancer in previous non-coding driver detection studies (*Melton et al., 2015*). The model considers four possible outcomes; transitions (TS$_{\{A\to G, \ G\to A\}}$), two types of transversions (TV$_{\{A\to T, \ G\to T\}}$ and TV$_{\{A\to C, \ G\to C\}}$) as well as the reference class of no mutation. The use of logistic regression ensures that the predicted probabilities are restricted to lie in the interval between zero and one. The reference sequence used in the model is the GRCh37 assembly (hg19) for the human genome. The explanatory variables of the model are listed below.

- Sample id: a factor variable with 505 levels.

- Replication timing: A numeric variable with values between zero (early replication) and one (late replication) (*Chen et al., 2010*). The variable is computed for 100 kb windows. Originally, the variable corresponds to the hg18 assembly for the human genome. It is converted to the hg19 assembly using the UCSC liftOver tool (*UCSC Genome Browser, 2016*).
- Trinucleotides: This variable is broken down into two separate variables; the reference bp in question as well as the left and right flanking bases. The bp in question is encoded as a factor variable with two levels, 'A' for A:T bps with weak hydrogen bonds, and 'G' for G:C bps with strong hydrogen bonds. The left and right flanking bases are implemented in a factor variable with 16 different levels, 'AA' to 'TT'. Including this variable in an interaction term with sample id effectively takes the sample-specific mutational signatures into account.
- Genomic segment: A factor variable with six levels, 'protein-coding genes', 'promoter elements', 'splice sites', '5' UTRs', '3' UTRs' and 'other'.
- Expression level: A numeric variable based on all available RNAseq expression data from TCGA (version 2, RSEM values, level 3 data). All RSEM values were $\log_2(x + 1)$ transformed. For each cancer type, the median expression was calculated for all genes. If multiple annotations of a gene existed, the longest annotation was used. For overlapping genes, the expression is summed up. We collapsed colon (COAD) and rectal carcinoma (READ) to a joint cancer-type CRC by averaging over the expression values (*Fredriksson et al., 2014*).
- Local mutation rate: A numeric variable calculated per base position. For each position, the position itself plus the flanking 10 kb on either side is skipped to avoid that the rate of the tested element has a large effect on local mutation rate. The local mutation rate is then based on the next flanking 20 kb regions on either side of the skipped regions. For each sample, the number of mutations in the two 20 kb regions are weighted by the reciprocal of the total number of mutations in the sample. The value of the local mutation rate is the weighted sum of the mutations across all samples.

The multinomial logistic regression model fit is based on a so-called *count-table*. For the purpose of creating this data structure, the three numeric variables, replication timing, local mutation rate, and expression level are each discretized into five bins. For each combination of explanatory variable levels ($505 \times 5 \times 2 \times 16 \times 6 \times 5 \times 5 = 12,120,000$ combinations), the number of genomic positions as well as the number of mutations of each type are counted. Before constructing the count-table, all COSMIC genes were excluded from the data set.

For fast and memory-efficient estimation, the multinomial logistic regression model is split up into three binomial logistic models (*Begg and Gray, 1984*). Estimation is conducted in R (*R Development Core Team, 2008*) (RRID:SCR_001905) using the function *glm4* from the contributed package *MatrixModels* (*Bates and Maechler, 2015*), which provides efficient estimation for GLMs with sparse design matrices. Three-fold multiple imputation is used to handle missing values in the variable replication timing (*Schafer, 1997*). The imputed values are randomly drawn from the marginal distribution of observed replication timing values.

## Scoring schemes

In the implementation of ncdDetect, the scores must be discrete values. For speed efficiency, integer values are recommended. The conservation scoring scheme is based on the phyloP scores. To ensure that all scores are positive, the phyloP values are shifted by adding 20 to all values. For computational reasons, the values are rounded downwards to the first decimal point, and multiplied by a factor of 10 to create integers. No mutation is associated with a score value of zero. The log-likelihood scoring scheme defines the scores as minus the natural logarithm of the sample- and position-specific neutral somatic mutation probabilities predicted by the null model. The scores are converted into integer values by the same procedure used for the conservation scores. Effectively, this means that positions with no mutations will be given a score of zero.

## Candidate elements

The candidate elements are defined based on the protein-coding transcript annotations of GENCODE version 19 basic annotation set (*GENCODE, 2016*). Regions are divided into five categories; protein-coding genes, promoter elements, splice sites, 3' UTRs, and 5' UTRs. The different regions are defined per transcript and collapsed per gene. Splice site regions are defined as the two intronic bases on either side of all internal exons. Promoter elements are defined as 500 bps in either direction from transcriptional start sites. A hierarchy for the categories is defined as protein-coding genes

> splice sites > 3' UTRs > 5' UTRs > promoter elements. Bps included in two or more categories are retained only for the category higher in the hierarchy. Region definitions are available in *Supplementary file 4*.

Elements located on chromosome X and Y are not considered in the analyses conducted here. The number of analyzed elements thus differ from the total number of elements defined (*Table 1*). We analyze 19,256 protein-coding regions, 19,157 promoter elements, 17,867 splice sites, 18,481 3' UTRs and 18,220 5' UTRs. The length distributions of the element types are depicted in *Figure 5— figure supplement 3*.

## An overdispersion-based mutation rate adjustment

To correct for overdispersion of the mutation rate, we adjust each sample- and position-specific mutational probability by an overdispersion-based mutation rate correction factor. The correction factor is modeled with a beta binomial model. Details are given in Appendix, section 1.

## TCGA exome data

To support our findings from the 505 whole genome TCGA samples, we obtain mutation calls, expression values and survival data based on the larger set of TCGA exomes (*Weinstein et al., 2013*; *Supplementary file 7*). These data are applied in the expression analyses, in the analyses of enrichment of G→A mutations in *SMUG1* mutated samples, as well as in the survival analyses performed.

We obtain mutation calls for 5802 samples. Of these, we remove 348 samples, which are also present in the original 505-sample data set. The final mutation call set thus consists of 5454 TCGA exome samples. We obtain expression data for 8471 samples, and after removing samples present in the 505-sample set, a total of 4295 samples have both mutation calls and expression data available. A total of 5336 TCGA exome samples have both mutation calls and clinical survival data available, after subtracting samples that are also present in the 505 whole genome TCGA sample set.

## Expression analysis

For a given candidate, we perform a two-sided Wilcoxon rank sum test. With this, we test the hypothesis that there is no difference in gene expression levels between samples that are mutated and samples that are not mutated in a given candidate element. Such a test is performed for each individual cancer type, and the p-values are combined across cancer types using Fisher's method.

These analyses are performed for both the 505 whole genome TCGA samples and the 4295 TCGA exome samples with the necessary data available. Data overview and results are available in *Supplementary file 5*.

## Enrichment of G→A mutations in *SMUG1* mutated samples

In order to test if the proportion of G→A (including C→T) mutations outside CpG sites are greater for samples harboring a *SMUG1* mutation, compared to samples not carrying such a mutation, we first count the 192 (=4 × 4 × 4 × 3) different mutation types (including left and right flanking bases for a given mutated position) for each of the samples. For each sample, we count the number of G→A mutations, excluding those in CpG sites and, for melanoma samples, those part of the CC→TT UV induced mutational signature (*Alexandrov and Stratton, 2014*). The counts are normalized by the total number of mutations for the sample. These proportions are referred to as the *uracil-DNA glycosylase deficiency signature*. For a given cancer type, we divide the samples into two groups; those that carry a *SMUG1* mutation, and those that do not. A one-sided Wilcoxon rank sum test is performed to test the null hypothesis that *SMUG1*-mutated samples do not have higher values of the uracil-DNA glycosylase deficiency signature statistic, compared to samples without *SMUG1* mutations. Fisher's method is used to combine p-values across cancer types. This type of analysis is performed for the 505 whole genome TCGA sample set.

We further analyze whether there is a correlation between the uracil-DNA glycosylase deficiency signature and *SMUG1*, *UNG*, or *SMUG1* × *UNG* gene expression. This type of analysis is performed for both the 505 whole genome TCGA samples and the 4295 TCGA exome samples with the necessary data available. The uracil-DNA glycosylase deficiency signature statistic calculated on the basis

of the TCGA exome data set is based only on captured CDS regions. Results and data overview are available in *Supplementary file 5*.

## Survival analysis

We investigate the correlation between mutation status and survival data in the following manner: We download clinical data from the TCGA data portal (*TCGA Data Portal, 2016*) (running date 01/11/2015) using the RTCGAToolbox R library (*Samur, 2014*). For a candidate driver element, the difference in survival between mutated and non-mutated samples is tested per cancer type using a one-sided Log-rank test on the Kaplan-Meier estimated survival curves (*Kaplan and Meier, 1958*). The tests are only performed when at least four mutations are observed within a given cancer type. Evidence is combined across cancer types with Fisher's method.

The survival analysis is performed for each of the top-50 ranked ncdDetect candidates of each non-coding element type (promoters, splice sites, 3' UTRs and 5' UTRs), or all significant elements of a given type. The analyses are performed for both the 505 whole genome TCGA sample set, and the 5336 TCGA exome sample set with mutation calls and clinical survival data available. Results and data overview are available in *Supplementary file 6*.

## Time complexity

The use of mathematical convolution on the fine grained sample- and position-specific scores and probabilities makes ncdDetect computationally intensive (*Figure 3—figure supplement 1*). Convolution is the procedure of calculating the distribution function of the sum of independent discrete random variables. The algorithm is implemented using dynamic programming (*Touzet and Varré, 2007*) and can be thought of as filling out a matrix from the bottom left corner to the upper right corner. Convoluting each cell in the matrix has time complexity $O(1)$, and the running time is thus determined by the size of the matrix. The time complexity of the algorithm is $O(m \times k \times s_{max})$, where m is the element size, k is the number of samples and $s_{max}$ is the maximum score.

## Implementation

ncdDetect is implemented in the software environment R (*R Development Core Team, 2008*) (RRID:SCR_001905), using the *Rcpp* and *RcppArmadillo* packages (*Eddelbuettel, 2016*) for speed optimization. The core ncdDetect functions to perform convolution are collected in the R package *ncdDetectTools* available at github.com. The package can be installed using the *devtools* package (*Tools to Make Developing R Packages Easier, 2016*): *install_github('MaleneJuul/ncdDetectTools')*. A few examples of the package functionalities are given in the package vignette, also available in the github repository.

The null model estimates used for the current application is provided at *http://moma.ki.au.dk/ncddetect/* along with a tutorial on how to obtain p-values from these estimates using ncdDetect.

## Availability of data and materials

Mutational, expression and clinical data for the TCGA samples are administered by dbGaP (https://dbgap.ncbi.nlm.nih.gov) (RRID:SCR_002709). The additional datasets supporting the conclusions of this article are included within the article and its supplementary files.

## Acknowledgements

We thank the TCGA consortium for data access and the system administrators of the GenomeDK high performance computing facility for facilitating the computational analysis.

## Additional information

### Funding

| Funder | Grant reference number | Author |
| --- | --- | --- |
| Medical Sciences | Sapere Aude Grant,#12-126439 | Jakob Skou Pedersen |

| The Danish Council for Strategic Research | #10-092320/DSF | Jakob Skou Pedersen |

The funders had no role in study design, data collection and interpretation, or the decision to submit the work for publication.

## Author contributions

MJ, Data curation, Software, Formal analysis, Visualization, Writing—original draft, Writing—review and editing, Developed, implemented and applied the ncdDetect algorithm and performed downstream analyses of results, Wrote the manuscript; JB, Data curation, Formal analysis, Writing—review and editing, Developed and implemented the statistical null model and provided the predicted mutational probabilities used in ncdDetect, Assisted with preparing the manuscript; QG, Data curation, Formal analysis, Developed and implemented the statistical null model and provided the predicted mutational probabilities used in ncdDetect; MMN, Data curation, Formal analysis, Provided genomic elements and contributed expression analysis, Helped visualise perform downstream analysis of the candidate elements, Assisted with preparing the manuscript; MŚ, Data curation, Formal analysis, Visualization, Provided correlation analysis of survival data and mutational status for candidate elements; HH, Data curation, Formal analysis, Visualization, Methodology, Participated in visualising and performing downstream analysis of the candidate elements, Further helped to curate the analysed tumor samples; TM, Software, Formal analysis, Methodology, Implemented the overdispersion-based mutation rate correction factor, and provided feedback and insights into dynamic programming and the use of the R-package Rcpp; AH, Conceptualization, Supervision, Writing—review and editing, Conceived the research project and supervised the development of ncdDetect, Revised final version of the manuscript; JSP, Conceptualization, Supervision, Funding acquisition, Visualization, Writing—original draft, Writing—review and editing, Conceived the research project and supervised the development of ncdDetect, Wrote the manuscript

## Author ORCIDs

Malene Juul, http://orcid.org/0000-0001-9722-0461
Jakob Skou Pedersen, http://orcid.org/0000-0002-7236-4001

# Additional files

## Supplementary files

• Supplementary file 1. Supplementary results. P-values obtained with ncdDetect using conservation scores.

• Supplementary file 2. Supplementary results. P-values obtained with ncdDetect using log likelihood scores.

• Supplementary file 3. Supplementary results. P-values obtained with ncdDetect using number of mutations as scores.

• Supplementary file 4. Region definitions. Definitions of candidate elements: Promoter regions, splice sites, 3' UTRs, 5' UTRs and protein-coding genes.

• Supplementary file 5. Expression analyses and a uracil-DNA glycosylase deficiency signature statistic. General expression analyses results and analyses performed to investigate the impact of SMUG1 mutations on expression levels as well as the uracil-DNA glycosylase deficiency signature statistic.

• Supplementary file 6. Correlation between mutation status and survival. Data overview and results obtained from the survival analyses.

• Supplementary file 7. Information on how to access expression and survival TCGA data sets. Overview of the specific TCGA samples included in the expression and survival analyses.

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

## Appendix

### Overdispersion-based rate adjustment

As our understanding of the mutational process is limited and as we do not know all relevant explanatory variables for all our samples, there will always be a difference between the predicted and actual mutation rate (*Appendix 1—figure 1*). The unaccounted for explanatory variables are likely to have auto-correlated regional effects. The effect of differences between actual and predicted mutation rates will thus be accumulated along elements and be most pronounced for long elements (*Appendix 1—figure 2*). Even small biases in the predicted versus actual mutation rate may become significant if elements are sufficiently long. In our case, the protein-coding genes are the longest element type and therefore the most likely to be affected by such biases.

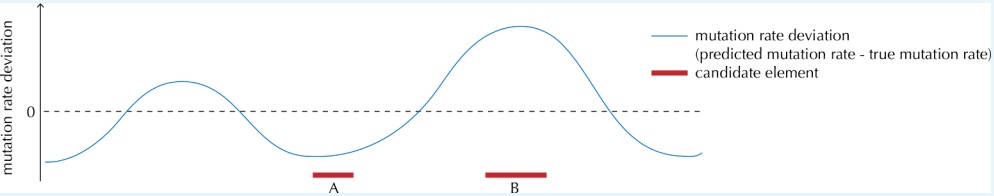

**Appendix 1—figure 1.** Illustration of the motivation behind the overdispersion-based rate adjustment. For candidate element A, we overestimate the mutation rate, and thus end up with a conservative p-value for this element when analysing it with ncdDetect. For candidate element B, on the other hand, we underestimate the mutation rate. In this case, ncdDetect will produce a p-value that is too small, creating a potential false-positive call. The effect of underestimating the mutation rate will be greater for longer candidate elements.

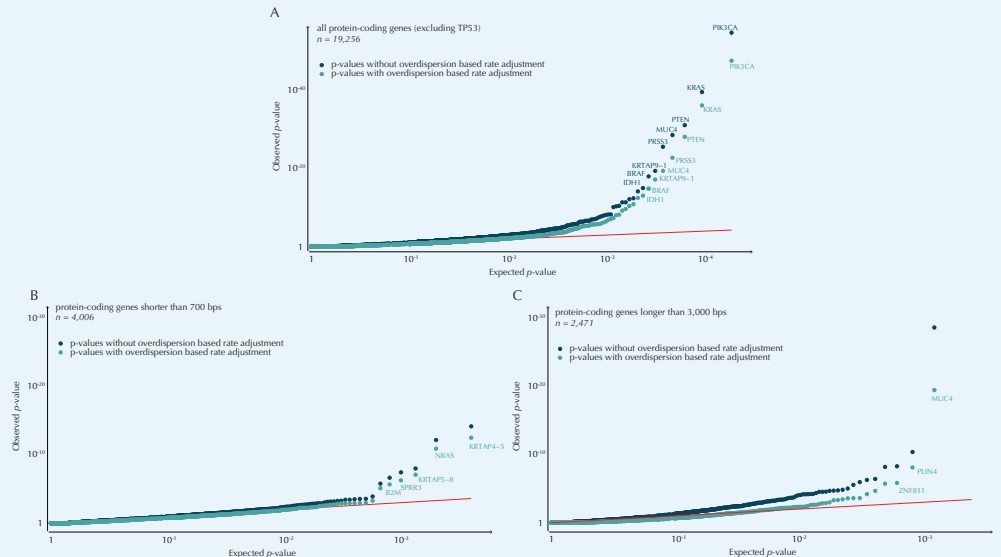

**Appendix 1—figure 2.** QQ-plots of p-values obtained with and without the overdispersion-based rate adjustment. (**A**) QQ-plots of all protein-coding genes (excluding *TP53* for illustration purposes). (**B**) QQ-plots of protein-coding genes shorter than 700 bp. For the shorter genes, the p-values are not particularly inflated. The overdispersion-based rate adjustment does not affect the distribution of p-values much. (**C**) QQ-plots of protein-coding genes longer than 3000 kb. For the longer genes, the p-values are inflated, and the overdispersion-based rate adjustment effectively corrects for much of this inflation.

**Appendix 1—figure 2—source data 1.** P-value and gene length for each protein-coding gene. The p-values are obtained with and without the overdispersion-based rate adjustment. This data set relates to *Appendix 1—figure 2*.

The unavoidable difference between actual and predicted mutation rates across elements and samples will increase the unexplained variance and lead to an overdispersion of the number of mutations per element (or other test statistics based on it). By capturing and taking this overdispersion into account, the specificity of the method can be improved, though not the power, which depends on reducing the unexplained variance by better mutational null models. To correct for overdispersion, we adjust each sample- and position-specific mutational probability by an overdispersion-based mutation rate correction factor.

The overdispersion-based rate adjustment is modelled with a beta binomial model. For a region of length $L_i$ having $X_i$ mutations, we have

$$X_i \sim \mathrm{Binomial}(L_i \cdot N_i, p_i),$$

$$p_i \sim \mathrm{Beta}(\alpha_i, \beta_i).$$

The parameters $\alpha_i$ and $\beta_i$ are constrained to satisfy that the expected mutation probability $\mu_i = \mathbb{E}[p_i] = \frac{\alpha_i}{\alpha_i + \beta_i}$ equals $\hat{p}_i$, the average mutation rate in the region predicted by the logistic regression model. Left with one degree of freedom the overdispersion is modelled with the parameter $\gamma = \frac{\mathbb{SD}(p)}{\mathbb{E}(p)} = \sqrt{\frac{\beta}{\alpha} \frac{1}{\alpha + \beta + 1}}$. We can express $\alpha$ and $\beta$ in terms of $\mu$ and $\gamma$:

$$\alpha = \frac{1 - \mu - \gamma^2 \mu}{\gamma^2},$$

$$\beta = \frac{1 - \mu}{\mu} \cdot \frac{1 - \mu - \gamma^2 \mu}{\gamma^2}.$$

In this alternative parameterization our model becomes

$$X_i \sim \mathrm{Binomial}(L_i \cdot N_i, p_i),$$

$$p_i \sim \mathrm{Beta}(\hat{p}_i, \gamma).$$

The parameter $\gamma$ is shared across all regions, and we estimate it by numerically maximizing the likelihood function

$$L(\gamma) = \prod_{i=1}^{K} \binom{NL_i}{X_i} \frac{B(X_i + \alpha, NL_i - X_i + \beta)}{B(\alpha, \beta)} = \prod_{i=1}^{K} \binom{NL_i}{X_i} \frac{B\left(X_i + \frac{1 - \mu - \gamma^2 \mu}{\gamma^2}, NL_i - X_i + \frac{1 - \mu}{\mu} \cdot \frac{1 - \mu - \gamma^2 \mu}{\gamma^2}\right)}{B\left(\frac{1 - \mu - \gamma^2 \mu}{\gamma^2}, \frac{1 - \mu}{\mu} \cdot \frac{1 - \mu - \gamma^2 \mu}{\gamma^2}\right)},$$

where B is the beta function. To avoid that regions under positive selection affect the estimate of $\gamma$, we filter out the top 5% and bottom 5% of regions, where the observed number of mutations deviates the most from the expected number of mutations. For protein-coding genes we further explicitly filter out COSMIC genes.

Inspecting the QQ-plot of the p-values for protein-coding genes shows that the overdispersion-based rate adjustment improves the overall fit of the p-values to uniformity

under the null and reduces the inflation of the tail of the distribution (*Appendix 1—figure 2A*). For short genes (<700 bp), the QQ-plots show a near perfect fit of the p-values to the uniform expectation with known or likely cancer drivers standing out (*Appendix 1—figure 2B*). For long genes (>3000 bp), the fit is much improved by the rate-adjustment, though some inflation is still present, resulting in significant calls that are likely false positives (e.g. *MUC4*, *PLIN4*, etc.) (*Appendix 1—figure 2C*).

# ncdDetect compared to other non-coding cancer driver detection methods

In order to benchmark the performance of ncdDetect, we compared our results to those obtained with two other non-coding cancer driver detection methods, ExInAtor (*Lanzós et al., 2017*) and LARVA (*Lochovsky et al., 2015*). ExInAtor is designed for the analysis of lncRNAs but is also applicable on protein-coding genes. We thus compared ncdDetect and ExInAtor on protein-coding genes. At the time of writing, LARVA does not support discontiguous element types (e.g. the joint analysis of multiple exons within a single gene). The promoter elements analyzed in the present paper are in principle contiguous. However, as we subtract overlapping annotations from 5' UTRs, they can be discontiguous in some cases. We have run LARVA on our promoter definitions, without removing any overlapping annotations from other element types.

The benchmarking on protein-coding genes is performed using the COSMIC Cancer Gene Census as a true-positive set (*Forbes et al., 2015*). The promoter benchmarking is not as straightforward, given the lack of a true positive set for this element type. We compare the results using previously described promoter cancer driver candidates.

## Performance on protein-coding genes

Where ncdDetect has a tendency to suffer from a significant fraction of false positive predictions, ExInAtor has a tendency to suffer from a significant fraction of false negative predictions. The published ExInAtor results on protein-coding genes, based on the same 505 cancer samples analyzed here, contain p-values for 19,309 protein-coding genes, where three are significant (q < 0.10). One of those three significant genes is in the COSMIC database. For ncdDetect, 64 genes are called significant (q < 0.10), of which 15 ($\approx 23\%$) are COSMIC genes. Notably, six of the top-ten ncdDetect candidates are COSMIC genes ($p = 2.0 \cdot 10^{-7}$, Fisher's exact test). This is the case for three of the top-ten ExInAtor candidates ($p = 3.3 \cdot 10^{-3}$, Fisher's exact test). In general, ExInAtor predicts much fewer candidates than ncdDetect, and thus have a lower false-positive rate. However, ncdDetect has a higher COSMIC Census recall rate, that is, it performs better at ranking genes compared to ExInAtor (*Appendix 1—figure 3*).

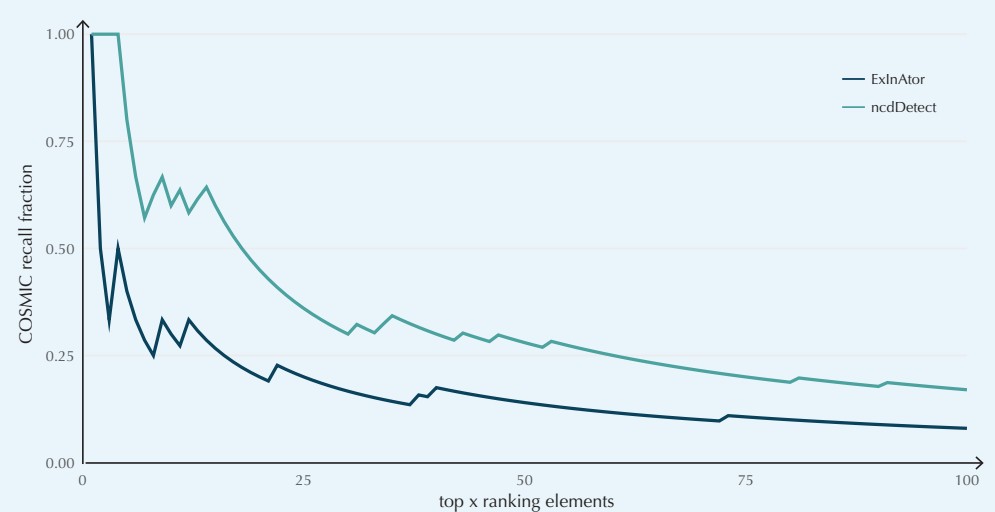

**Appendix 1—figure 3.** COSMIC Gene Census recall plot. The fraction of COSMIC genes recalled in the top ncdDetect and ExInAtor candidates.

**Appendix 1—figure 3—source data 1.** COSMIC Gene Census recall data. The fraction of recalled COSMIC genes in the top ncdDetect and ExInAtor candidates. This data set relates to *Appendix 1—figure 3*.

Looking closer at the top-15 ncdDetect protein-coding candidates, we find that nine are COSMIC genes (*Appendix 1—table 1*). Non-COSMIC genes in the list include *PRSS3*, *KRTAP9-1*, *KRTAP4-5* and *BCLAF1*. All these genes have some reported cancer association: The expression of *PRSS3* has been shown to be upregulated in metastatic prostate cancer and is also associated to pancreatic and lung cancer (*Jiang et al., 2010*; *Hockla et al., 2012*; *Marsit et al., 2005*). Although the Keratin-associated proteins *KRTAP9-1* and *KRTAP4-5* have no wide spread reported role in cancer, a recent study found that they can play a part in malignant progression (*Berens et al., 2017*). Finally, *BCLAF* has been associated to colon cancer (*Zhou et al., 2014*). The remaining two genes in the list, *MUC4* and *AL390778.1* have no reported cancer driver potential. Interestingly, *MUC4* continues to be significant in our analyses, even after overdispersion-based rate adjustment.

**Appendix 1—table 1.** Analysis of the top 15 ncdDetect protein-coding candidates.

| Rank | Gene name | q-value | Size (bp) | Cosmic | Conclusion |
|---|---|---|---|---|---|
| 1 | TP53 | $1.15 \times 10^{-235}$ | 1378 | True | Putative cancer driver gene |
| 2 | PIK3CA | $2.82 \times 10^{-44}$ | 3207 | True | Putative cancer driver gene |
| 3 | KRAS | $4.93 \times 10^{-33}$ | 708 | True | Putative cancer driver gene |
| 4 | PTEN | $3.70 \times 10^{-25}$ | 1212 | True | Putative cancer driver gene |
| 5 | PRSS3 | $6.80 \times 10^{-20}$ | 1056 | False | Reported cancer association |
| 6 | MUC4 | $1.26 \times 10^{-16}$ | 16,239 | False | Likely false positive due to length |
| 7 | KRTAP9-1 | $1.77 \times 10^{-14}$ | 770 | False | Reported cancer association |
| 8 | BRAF | $3.25 \times 10^{-12}$ | 2301 | True | Putative cancer driver gene |
| 9 | IDH1 | $1.76 \times 10^{-10}$ | 1245 | True | Putative cancer driver gene |
| 10 | KRTAP4-5 | $6.11 \times 10^{-10}$ | 546 | False | Reported cancer association |
| 11 | NRAS | $2.20 \times 10^{-8}$ | 570 | True | Putative cancer driver gene |
| 12 | AL390778.1 | $5.95 \times 10^{-8}$ | 735 | False | No reported cancer driver properties |

*Appendix 1—table 1 continued on next page*

*Appendix 1—table 1 continued*

| Rank | Gene name | q-value | Size (bp) | Cosmic | Conclusion |
|------|-----------|---------|-----------|--------|------------|
| 13 | NFE2L2 | $3.15 \times 10^{-7}$ | 1890 | True | Putative cancer driver gene |
| 14 | FBXW7 | $7.35 \times 10^{-7}$ | 2618 | True | Putative cancer driver gene |
| 15 | BCLAF1 | $7.92 \times 10^{-6}$ | 2763 | False | Reported cancer association |

## Performance on non-coding regulatory elements

The LARVA analysis of the 20,052 defined promoter elements yields 16 significant candidates (q < 0.10). ncdDetect agrees, and calls all of these 16 promoters significant, along with an additional 144 candidates (q < 0.10). Several of the cases detected by ncdDetect, and not by LARVA, have previously been described to be associated with cancer. These include *PLEKHS1* and *WDR74* as described in the main text. Further, promoter mutations in *DPH3* and *OXNAD1* have been associated to skin cancers (*Denisova et al., 2015*). A number of the ncdDetect identified candidates are also identified in an earlier cancer study (*Weinhold et al., 2014*), including *SMUG1* (*Appendix 1—figure 4*). Finally, the protein-coding genes associated to the promoters *KDM5A*, *CNOT3* and *NCOR1* are COSMIC genes, and detected solely by ncdDetect. The promoter region of *PRSS3*, a case study in the main text, is also detected by ncdDetect alone. Taken together, several of the ncdDetect promoter candidates that are not detected by LARVA have previously reported cancer driver potential.

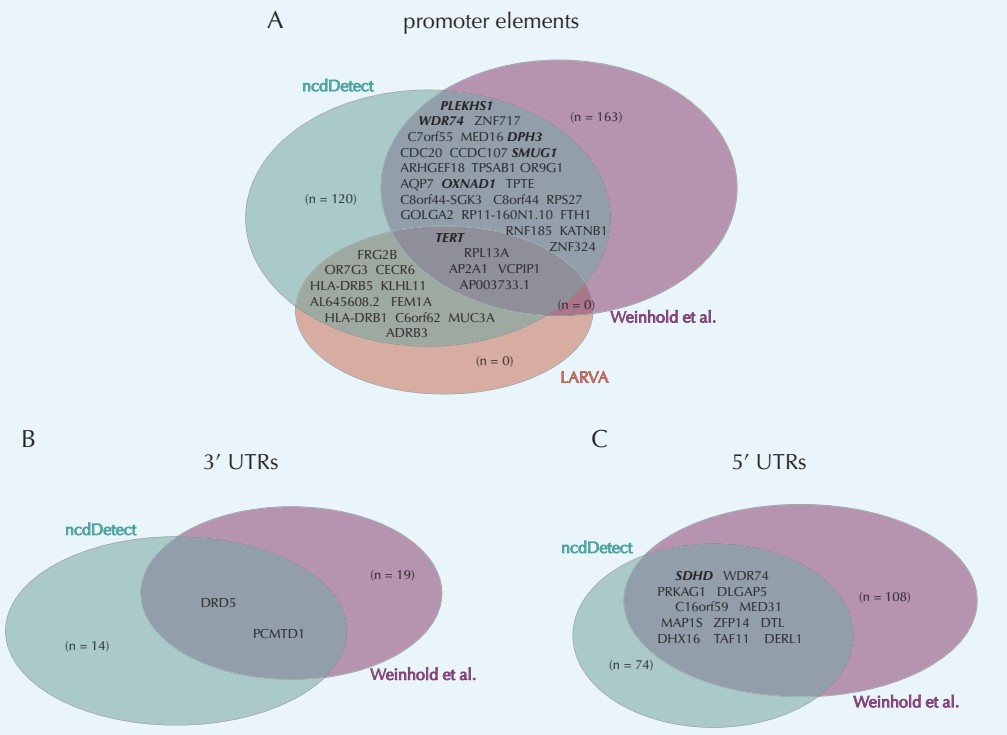

**Appendix 1—figure 4.** Illustration of overlap between significant elements found by ncdDetect and other non-coding cancer driver screens. Highlighted elements are mentioned in the text. (**A**) Overlap of promoter elements found to be significant with ncdDetect and LARVA, as well as promoter elements previously described in a non-coding cancer driver screen (*Weinhold et al., 2014*). We note that *TERT* and *PLEKHS1* are also detected by a second non-coding driver screen (*Melton et al., 2015*). (**B**) Overlap between 3' UTRs detected by ncdDetect and 3' UTRs detected by a previous study (*Weinhold et al., 2014*). (**C**) Overlap

between 5' UTRs detected by ncdDetect and 5' UTRs detected by a previous study (*Weinhold et al., 2014*). We note, that out of the 863 whole genomes analyzed in (*Weinhold et al., 2014*), 356 are sequenced by the TCGA. These samples appear to be a subset of the 505 TCGA samples analyzed here. The data sets are thus not completely independent.

## Algorithmic details of ncdDetect

Assume that a candidate element has $m$ positions and that somatic mutations are called for a total of $k$ samples. The sample- and position-specific probabilities of a mutation are predicted using the null model. The four outcomes of the model are one type of transition ($\mathrm{TS} = \mathrm{TS}_{\{A \to G, G \to A\}}$), two types of transversions ($\mathrm{TV}_1 = \mathrm{TV}_{\{A \to T, G \to T\}}$ and $\mathrm{TV}_2 = \mathrm{TV}_{\{A \to C, G \to C\}}$), as well as the reference class of no mutation ($\mathrm{NM}$). The corresponding probabilities for the $i$'th sample ($i = 1, \ldots, k$) and position $j$ ($j = 1, \ldots, m$) are (*Figure 3C*)

$$\left( \pi_{ij}^{\mathrm{TS}}, \pi_{ij}^{\mathrm{TV}_1}, \pi_{ij}^{\mathrm{TV}_2}, \pi_{ij}^{\mathrm{NM}} \right). \tag{1}$$

Associated to each outcome is a sample- and position-specific score (*Figure 3B*)

$$\left( \mathrm{score}_{ij}^{\mathrm{TS}}, \mathrm{score}_{ij}^{\mathrm{TV}_1}, \mathrm{score}_{ij}^{\mathrm{TV}_2}, \mathrm{score}_{ij}^{\mathrm{NM}} \right).$$

Let $\mathrm{obs}(i,j)$ indicate the observed outcome for position $i$ and sample $j$ (*Figure 3D*). Then the observed score for position $i$ and sample $j$ is $\mathrm{score}_{ij}^{\mathrm{obs}(i,j)}$, the cumulated observed sample-specific score is given by (*Figure 3E*)

$$\mathrm{score}_i^{obs} = \sum_{j=1}^{m} \mathrm{score}_{ij}^{\mathrm{obs}(i,j)},$$

and the overall score is (*Figure 3F*)

$$\mathrm{score}_{overall}^{obs} = \sum_{i=1}^{k} \mathrm{score}_i^{obs} = \sum_{i=1}^{k} \sum_{j=1}^{m} \mathrm{score}_{ij}^{\mathrm{obs}(i,j)}.$$

We now describe how to determine the null distribution for the overall score. First consider the null distribution for the sample-specific score (*Figure 3E*). Let $Z(i,j)$ be the stochastic variable that indicates the outcome for position $i$ and sample $j$. Each of the four outcomes ($TS$, $TV_1$, $TV_2$, $NM$) happen with the probability determined by *Equation (1)*. The cumulated sample-specific score distribution is thus the distribution of the stochastic variable

$$A_{i,m} = \sum_{j=1}^{m} \sum_{z \in \{\mathrm{TS}, \mathrm{TV}_1, \mathrm{TV}_2, \mathrm{NM}\}} \mathrm{score}_{ij}^{z} \mathbb{1}(Z(i,j) = z),$$

where $\mathbb{1}(\cdot)$ is the indicator function. If we assume that scores are non-negative integers we have the recursion from one position to the next

$$P\big(A_{i,j} = s\big) = \sum_{\ell=0}^{s} P\big(A_{i,(j-1)} = \ell\big) \sum_{z \in \{\text{TS},\text{TV}_1,\text{TV}_2,\text{NM}\}} P\Big(\text{score}_{i,j}^{z} = s - \ell\Big).$$

Second consider the null distribution for the overall score (**Figure 3G**)

$$B_k = \sum_{i=1}^{k} A_{i,m}.$$

A similar recursion as before holds for the overall score distribution. We can include the next sample from the recursion

$$P(B_k = s) = \sum_{\ell=0}^{s} P(B_{k-1} = \ell) P\big(A_{k,m} = s - \ell\big).$$

The final p-value for the element of interest is

$$P\big(B_k \geq \text{score}_{overall}^{obs}\big) = 1 - P\big(B_k < \text{score}_{overall}^{obs}\big).$$

