## [Decision Letter]

Thank you for submitting your article "Non-coding cancer driver candidates identified with a sample- and position-specific model of the somatic mutation rate" for consideration by *eLife*. Your article has been favorably evaluated by Aviv Regev (Senior Editor) and two reviewers, one of whom is a member of our Board of Reviewing Editors. The following individual involved in review of your submission has agreed to reveal his identity: David C Wedge (Reviewer #2).

The reviewers have discussed the reviews with one another and the Reviewing Editor has drafted this decision to help you prepare a revised submission.

Summary:

This paper introduces a statistical method for detecting non-coding driver mutations in cancer data. It is applied to a modest pan-cancer genome data set (505 tumour-normals). The method recovers some known driver mutations (both coding and non-coding) and identifies a number of other candidates, some of which have corroborative evidence from gene expression and/or survival data. The identification of non-coding driver mutations is an important and unsolved problem that will be of growing importance as whole genome sequence data sets accumulate.

Essential revisions:

1) The method needs to be compared to other approaches (e.g. the Gerstein group approach) for identifying non-coding driver mutations. Given the computational cost of the new approach, it is important to demonstrate that it leads to a substantial advance in precision compared to simpler methods.

2) The Bertl et al. paper referred to as in preparation that describes the method needs to be made accessible – e.g. on a preprint site. Without this, important aspects of the approach cannot be fully assessed. Ideally, this would be combined with the current paper to make a coherent piece of work.

3) The COSMIC protein-coding driver mutation analysis is worrying. It suggests that at q=0.25, only 16% of driver mutations identified are in COSMIC. Given the relatively small size of the dataset used here (compared to the very large exome data sets represented in COSMIC), the 84% of 'drivers' that are not known need to be investigated further. E.g. are there additional mutation-rate varying factors that are relevant for protein-coding mutations?

4) Mutation rate is strongly affected by chromatin state (e.g. Polak, Nature Biotech, 2014) and this should be added as a co-variate. Further, it's unclear whether the model takes into account the sample-specific mutational signature. For example, the probability of TCW→TTW and TCW→TGW mutations is much greater in samples prone to APOBEC-induced mutational processes, particularly in regions of kataegis.

5) It is unclear how the scoring systems are used within the model. The probability of each mutation under a null model seems to be calculated without using the scores and the Q-Q analysis is conducted without apparent use of the scores, so it is not clear how 3 different sets of significant genes are identified.

---

## [Author Response]

*Essential revisions:*

*1) The method needs to be compared to other approaches (e.g. the Gerstein group approach) for identifying non-coding driver mutations. Given the computational cost of the new approach, it is important to demonstrate that it leads to a substantial advance in precision compared to simpler methods.*

We agree that comparison to existing methods is appropriate, although performance comparison is challenging given the sparsity of known non-coding cancer drivers. A limited number of non-coding driver screens have been published and the underlying methods are often not accessible. We originally compared against the results of published non-coding driver screens (Weinhold et al. 2014; Melton et al. 2015) (Figure 4). Although their data sets overlap the one we use, they are not identical.

We have identified three additional non-coding driver detection methods to compare ncdDetect against: OncoDriveFML (Mularoni et al. 2016), ExInAtor (Lanzós et al. 2017), and LARVA (Lochovsky et al. 2015). There are, however, complications involved with this task, as described below.

OncoDriveFML: The method has a web-interface (http://bg.upf.edu/oncodrivefml/analysis) that requires the user to upload the mutational catalogue to analyse. Given that the 505 samples of this study are restricted access, this is not possible.

ExInAtor: A method originally designed for lncRNAs, which can also be applied to protein-coding genes. Adapting the method to the region type elements we are working with (promoters, UTRs, and splice sites) is not straightforward, but we are able to compare the performance of ncdDetect and ExInAtor on protein-coding genes.

LARVA: This non-coding driver detection method is available as a Docker Image, and we are able to run the method with our data. However, after correspondence with the authors behind LARVA, it is clear that the current release of LARVA does not support discontiguous regions (e.g. the combined analysis of multiple exons within a gene, or the joint analysis of multiple splice sites). The promoter elements analysed here are in principle contiguous. However, as we subtract overlapping annotations from 5’ UTRs, they can be discontiguous in some cases. We have thus run LARVA on our promoter definitions, without removing any overlapping annotations from other element types.

The comparison between ncdDetect, LARVA and ExInAtor is added to the Appendix, section 2, and is described in the main text, subsection “Recall of known protein-coding drivers”, last paragraph and subsection “Recall and function of previously described non-coding drivers”, first paragraph.

*2) The Bertl et al. paper referred to as in preparation that describes the method needs to be made accessible – e.g. on a preprint site. Without this, important aspects of the approach cannot be fully assessed. Ideally, this would be combined with the current paper to make a coherent piece of work.*

The Bertl et al. paper and the present work are independent contributions. The Bertl et al. paper has as its main focus to understand the neutral mutational processes in cancer. In particular, the main aim of the Bertl et al. paper is to quantify how the heterogeneity in the mutation rate can be explained in terms of various explanatory variables such as cancer type, replication timing and expression level. One possible application of this work is cancer driver detection, as described in the present paper. The two papers are thus closely related, but they are concerned with two separate questions (Bertl et al: Model choice and selection for the neutral somatic mutation rate in cancer genomes; Juul et al: Driver detection), and constitute two separate contributions.

The Bertl et al. paper will be published on a preprint server within the next three weeks. We will provide the appropriate reference as soon as it is available.

*3) The COSMIC protein-coding driver mutation analysis is worrying. It suggests that at q=0.25, only 16% of driver mutations identified are in COSMIC. Given the relatively small size of the dataset used here (compared to the very large exome data sets represented in COSMIC), the 84% of 'drivers' that are not known need to be investigated further. E.g. are there additional mutation-rate varying factors that are relevant for protein-coding mutations?*

As our understanding of the mutational process is limited and as we do not know all relevant explanatory variables for all our samples, there will always be a difference between the predicted and actual mutation rate. The unaccounted for explanatory variables are likely to have auto-correlated regional effects. The effect of differences between actual and predicted mutation rates will thus be accumulated along elements and be most pronounced for long elements (Figure 9). Even small biases in the predicted versus actual mutation rate may become significant if elements are sufficiently long. In our case, the protein-coding genes are the longest element type and therefore the most likely to be affected by such biases.

If we could predict the actual mutation rate perfectly, we would have the most power to discriminate between actual driver elements and false positives. We therefore continue to strive to improve our model of site-specific mutational process.

The unavoidable difference between actual and predicted mutation rates across elements and samples will increase the unexplained variance and lead to an overdispersion of the number of mutations per element (or other test statistics based on it). By capturing and taking this overdispersion into account, the specificity of the method can be improved, though not the power, which depends on reducing the unexplained variance by better mutational null models (hence our long-term aim).

In the revised version of the paper, we correct for overdispersion of the number of mutations by adjusting the sample- and position specific mutation probabilities with an overdispersion-based rate adjustment. The theory and implementation of this approach is explained in detail in the Appendix, section 1. Analyses of the effect of the mutation rate correction are also presented in this section. The effect of overdispersion is greatest for longer candidate element, and is thus mostly pronounced for protein-coding genes.

Running ncdDetect with the updated model fit, including overdispersion-based rate adjustment, yields 64 protein-coding driver candidates, of which 15 (= 23%) are COSMIC genes. The results are thus improved from the first version of the paper, although there are still likely false positives in the candidate list.

When inspecting the QQ-plot of the (-log) p-values for protein-coding genes, it can be seen that the overdispersion-based rate adjustment improves the overall fit of the p-values to uniformity under the null and reduces the inflation of the tail of the distribution (Figure 9). For short genes (<700 bp), the QQ-plots show a near perfect fit of the p-values to the uniform expectation with known or likely cancer drivers standing out (Figure 9). For long genes (>3,000 bp), the fit is much improved by the rate adjustment, though some inflation is still present, resulting in significant calls that are likely false positives (e.g., MUC4, PLIN4, etc.; Figure 9).

We thus acknowledge that the false positive rate among long genes is probably not properly controlled and likely higher than the applied FDR threshold. We now acknowledge and extensively discuss these aspects in the manuscript: subsection “Model selection”, last paragraph, Discussion, third paragraph and subsection “Implementation”; Appendix, section 1). On the other hand, the performance on the short non-coding elements appears fine and unproblematic (Figure 5, Figure 5—figure supplement 1).

Although ncdDetect includes likely false positives in its candidate list, it performs well at ranking genes. For instance, six out of the top ten candidates are COSMIC genes (p = 2 x 10^-7^, Fisher’s exact test). A closer analysis of the top-15 protein-coding genes identified by ncdDetect shows that nine are COSMIC genes, and that four additional genes show cancer driver potential. Only two of the top-15 candidates do not have reported cancer association, one of them being MUC4, which is most likely a false positive given its long length. This analysis is added to the Appendix, section 2. A comparison of ncdDetect and ExInAtor on protein-coding genes showed that ncdDetect is superior in ranking genes, although the probable false-positive rate is elevated (Figure 10).

*4) Mutation rate is strongly affected by chromatin state (e.g. Polak, Nature Biotech, 2014) and this should be added as a co-variate. Further, it's unclear whether the model takes into account the sample-specific mutational signature. For example, the probability of TCW→TTW and TCW→TGW mutations is much greater in samples prone to APOBEC-induced mutational processes, particularly in regions of kataegis.*

It has been demonstrated that the mutation rate is highly affected by chromatin state, and in particular that the impact of chromatin structure on the mutation density is highly cell-type specific (Polak et al. 2015). However, we were not able to consistently include cell-type specific chromatin data as an explanatory variable in our model, as we could not identify relevant chromatin data corresponding to all the cancer types included in the present study.

Instead, we experimented with including tissue-agnostic DNaseI hypersensitivity as an explanatory variable in the null model. The details are described in the attached Bertl et al. paper, but the relevant figure and table on DNaseI hypersensitivity are included as Figure 12 and Table 2 for convenience.

The explanatory variables that go into the null model are selected using forward selection. The final model of Bertl et al. includes trinucleotide context, phyloP score, replication timing and tissue-specific expression level (Model 8 in the Figure 12 and Table 2). Adding more variables (including DNaseI hypersensitivity) did not improve the model fit as measured by deviance loss, as shown in Table 2. However, the analyses show that adding expression level as an explanatory variable in the model improved the fit significantly. We have thus updated the model fit used in the first version of this paper to also include the tissue-specific expression level. All analyses were repeated and the results presented in the revised version of the paper reflects this improved model fit. Note that the null model used for ncdDetect does not contain the explanatory variable phyloP, as it is used as a scoring scheme in the analyses.

Author response image 1.**DOI:**
http://dx.doi.org/10.7554/eLife.21778.049

Author response table 1.Table 3. Deviance loss and McFadden’s pseudo *R^2^* for each of the models tested in Step 3 to obtain model 8.**DOI:**
http://dx.doi.org/10.7554/eLife.21778.050ModelDeviance lossMcFadden’s pseudo R^2^Model 7 + βexpr.,cmut.typeχexpr.,c,i0.00032480.2258Model 7 + βelement,cmut.typeχelement,i0.00032790.2154Model 7 + βrepeat.,cmut.typeχrepeat.,i0.00032890.2154Model 7 + βCGcont.,cmut.typeχCGcont.,i0.00032890.2152Model 7 + βDNase1,cmut.typeχDNase1,i0.00032900.2150Model 7 + βCGI,cmut.typeχCGI,i0.00032910.2150

Regarding the sample-specific mutational signatures, these were and still are taken into account in the model. We include an interaction term between the variables sample id and trinucleotides, which contains information on the mutated base itself as well as the left and right flanking bases. This has now been clarified in the main text: subsection “Position- and sample-specific model of the background mutation rate”, last paragraph and subsection “Statistical null model”, first paragraph.

*5) It is unclear how the scoring systems are used within the model. The probability of each mutation under a null model seems to be calculated without using the scores and the Q-Q analysis is conducted without apparent use of the scores, so it is not clear how 3 different sets of significant genes are identified.*

It is true, that the sample- and position specific mutational probabilities are calculated without using the scores. They are based only on the explanatory variables included in the model (i.e. sample id, replication timing, trinucleotides, genomic segment, local mutation rate and expression level). However, the scores affect the test statistic and hence the significance as well as the QQ-plots.

As explained by Figure 3, ncdDetect takes four inputs: The candidate element to analyse, the somatic mutation calls, the sample- and position specific mutational probabilities as well as the sample-and position specific scores derived from the chosen scoring scheme.

The test statistic used in the significance evaluation performed by ncdDetect is the *observed score*. This value is defined as the sum of sample- and position-specific observed scores across the specific element that is being tested. For a given sample and a given position, the observed score will depend on the chosen scoring scheme. For instance, the scoring scheme using the number of mutations will always give a score of 1 to a mutated position, and a score of 0 to an unmutated position. The scoring scheme using phyloP will give a score corresponding to the position-specific phyloP value to a mutated position, and a score of 0 to an unmutated position. Thus, the observed score for a specific element will depend on the chosen scoring scheme.

The observed score is significance evaluated in the *background score distribution*. Again, the shape of this distribution will depend on the chosen scoring scheme: All possible sample- and position-specific scores for the chosen scoring scheme are convoluted together with the associated sample- and position-specific mutational probabilities to form the background score distribution.

In conclusion, the three different scoring schemes described in the text will produce three distinct sets of p-values. We have modified the text: subsection “ncdDetect significance evaluation”, as well as the legend of Figure 3, to make this point more clear.